# Demystifying When Pruning Works via Representation Hierarchies

Shwai He [1]   Guoheng Sun [1]   Haichao Zhang [2]   Yun Fu [2]   Ang Li [1]

## Abstract

Network pruning, which removes less important parameters or architectures, is often expected to improve efficiency while preserving performance. However, this expectation does not consistently hold across language tasks: pruned models can perform well on non-generative tasks but frequently fail in generative settings. To demystify how such discrepancies arise under pruning, we analyze network pruning from a representation-hierarchy perspective, decomposing the internal computation of language models into three sequential spaces: *embedding* (hidden representations), *logit* (pre-softmax outputs), and *probability* (post-softmax distributions). While representations in the embedding and logit spaces are largely robust to pruning-induced perturbations, the subsequent nonlinear transformation from logits to the probability space amplifies such deviations, whose persistence across time steps leads to substantial degradation during generation. By contrast, the stability of the categorical-token probability subspace, together with the robustness of the embedding space, supports the effectiveness of pruning for non-generative tasks such as retrieval and multiple-choice classification. Our representation-level analysis disentangles the effects of pruning across tasks and offers practical guidance for applying pruning effectively. The code is available in the project repository.

## 1. Introduction

Network pruning (Kusupati et al., 2020; Zhuang et al., 2020; Sun et al., 2024) is an effective approach for improving computational efficiency by removing less important parameters or architectures. As large language models continue to grow in scale (OpenAI, 2024; DeepSeek-AI, 2024; Team,

[1]University of Maryland, College Park, MD, USA [2]Northeastern University, Boston, MA, USA. Correspondence to: Ang Li <angliece@umd.edu>.

*Proceedings of the $43^{rd}$ International Conference on Machine Learning*, Seoul, South Korea. PMLR 306, 2026. Copyright 2026 by the author(s).

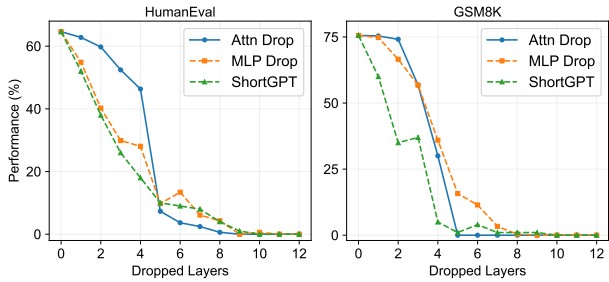

*(a)* Generative tasks.

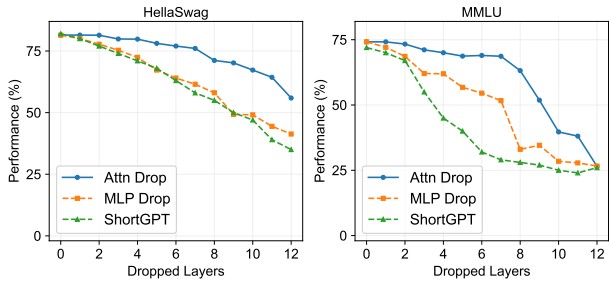

*(b)* Non-generative tasks.

*Figure 1.* **Effect of inter-layer pruning on generative and non-generative tasks**. Inter-layer pruning is implemented by removing entire transformer blocks (ShortGPT (Men et al., 2025)) or attention/MLP layers (Attn/MLP Drop (He et al., 2026)).

2025), compression via network pruning has become an increasingly attractive strategy for mitigating memory and computational costs.

However, as illustrated in Figure 1, the effectiveness of network pruning does not hold uniformly across language tasks (He et al., 2026). Empirically, pruned models often retain strong performance on non-generative tasks (Hendrycks et al., 2021; Zellers et al., 2019), which primarily depend on sequence-level representations or logits over a fixed set of categorical options, but frequently fail on generative tasks (Cobbe et al., 2021; Chen et al., 2021), where models generate output sequences by sampling from predicted probability distributions.

To investigate the root cause of this discrepancy, we analyze pruning from the perspective of internal representation transformations in language models. Specifically, we decompose

model computation along the inference pipeline into three sequential spaces: *embedding* (hidden representations), *logit* (pre-softmax outputs), and *probability* (post-softmax distributions). This decomposition naturally aligns with the distinct representational spaces involved in non-generative and generative tasks, while also providing a clear framework for tracing how pruning-induced perturbations propagate across different stages of the model and affect downstream performance.

Our empirical analyses reveal a clear representation hierarchy under pruning. The embedding space remains largely robust, exhibiting only minor deviations even when a substantial fraction of parameters is removed, consistent with prior findings (Gromov et al., 2025; He et al., 2026). Interestingly, the subsequent linear transformation from the embedding space to the logit space preserves comparable representational similarity.

In contrast, our empirical and theoretical analyses show that the nonlinear projection from logits to probabilities (Xuan et al., 2025) amplifies pruning-induced perturbations in the probability space, leading to disproportionately large deviations in the output distribution and ultimately destabilizing the generation process. These deviations persist across generation steps, further resulting in substantial degradation of generation quality. By contrast, non-generative tasks typically rely on the logits or probabilities of a small set of predefined option tokens at a single decision step, which remain comparatively stable under pruning. Together with the robustness of the embedding space, this property explains why network pruning remains effective for non-generative tasks such as retrieval and multiple-choice classification.

Through combined empirical and theoretical analyses, we develop a representation-level understanding of how pruning affects internal representations and why its impact differs across tasks. These findings explain why network pruning remains effective for non-generative tasks but poses substantial risks for generative ones, offering practical guidance for applying pruning. In summary, the contribution of this work is as follows:

- This work reveals a clear discrepancy in the effectiveness of network pruning across non-generative and generative tasks.

- For generative tasks, we identify the nonlinear mapping from logits to probabilities as a key mechanism that amplifies pruning-induced perturbations, leading to severe performance degradation.

- By contrast, low pruning-induced perturbations in the embedding and logit spaces, as well as the stability of the categorical-token probability subspace, support the effectiveness of network pruning in non-generative tasks and provide practical guidance for its application.

## 2. Related Works

**Efficiency Challenges in Large Language Models** Scaling large language models (LLMs) has driven rapid progress across a wide range of tasks, demonstrating strong and increasingly general capabilities (OpenAI, 2024; DeepSeek-AI, 2024; Team, 2025). However, such improvements often come at a substantial efficiency cost: the massive model parameters and the intermediate representations maintained during inference incur significant memory and computational overhead, posing challenges for real-time and resource-constrained deployment. As a result, how to trade off model capability and efficiency has become a central problem in modern LLM systems (Hoffmann et al., 2022; Wan et al., 2024). Importantly, language models exhibit fundamentally different inference behaviors between single-pass settings (e.g., one-step prefilling) and multi-step generation settings, suggesting that the effects of efficient methods like network pruning are inherently regime-dependent.

**Model Compression via Network Pruning** Network pruning, motivated by the substantial redundancy inherent in large language models, aims to reduce memory footprint and inference cost by removing less important components (Liu et al., 2019; Tanaka et al., 2020; Cheng et al., 2024; Zhang & Fu, 2025). Existing approaches can be broadly categorized into two classes: (i) unstructured weight sparsification (e.g., Wanda (Sun et al., 2024) and SparseGPT (Frantar & Alistarh, 2023)), and (ii) structured pruning of coupled structures such as layers or blocks (Gromov et al., 2025; He et al., 2026; 2025a; Zhang et al., 2025a). These pruning approaches primarily operate in the embedding space and have mainly been shown to succeed on non-generative tasks (Sun et al., 2024; Frantar & Alistarh, 2023; Lei et al., 2025; Zhang et al., 2025b; He et al., 2025b), which typically depend on the model's hidden representations or logits at a single inference step without iterative feedback across decoding steps. In contrast, generative tasks pose additional challenges for network pruning. For instance, errors introduced at earlier time steps can propagate to subsequent steps. In this work, we analyze how network pruning affects non-generative and generative tasks differently and uncover the underlying principles for effective pruning.

## 3. Background on Language Modeling

Modern language models process text by mapping discrete tokens to continuous representations, transforming them through multiple continuous latent spaces, and finally producing probability distributions over discrete tokens. Formally, given an input text sequence $\mathcal{T}$, the model first applies a tokenizer $\tau(\cdot)$ to map text into discrete tokens, i.e., $x = \tau(\mathcal{T})$ with $x_i \in \{1, \ldots, |\mathcal{V}|\}$, where $|\mathcal{V}|$ denotes the size of the vocabulary. Each token $x_i$ is then mapped to a

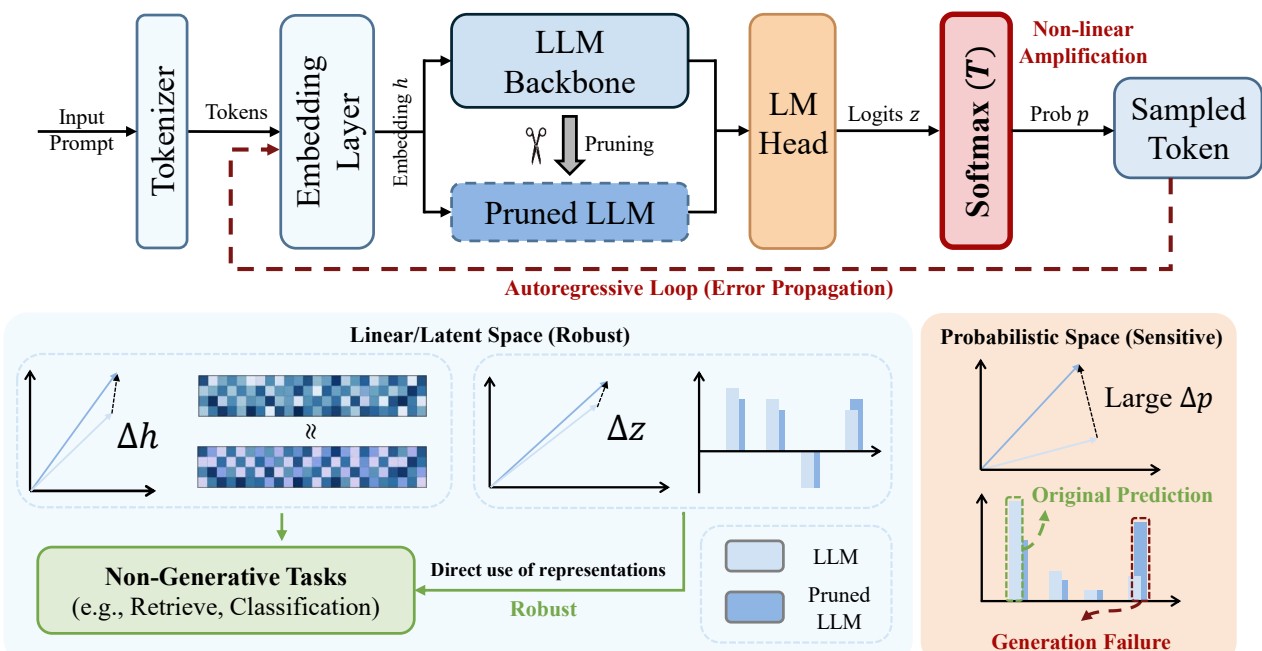

Figure 2. **Propagation of pruning-induced perturbations across representation spaces in LLMs.** Small embedding perturbations $\Delta h$ introduced by pruning remain stable in the logit space (i.e., small $\Delta z$), but are amplified by the softmax nonlinearity in the high-dimensional probability space, resulting in large probability shifts $\Delta p$ and degraded autoregressive generation.

continuous embedding vector through an embedding lookup table $\mathcal{E} \in \mathbb{R}^{|\mathcal{V}| \times d}$: $e = \mathcal{E}[x] \in \mathbb{R}^d$, where $d \ll |\mathcal{V}|$. The sequence of embeddings $e$ is processed by a deep neural network composed of $L$ layers, yielding a hierarchy of hidden representations:

$$h^{(l)} = f^{(l)}\left(h^{(l-1)}\right), \quad l = 1, \ldots, L, \qquad (1)$$

where $h^{(0)} = e$, and $f^{(l)}(\cdot)$ denotes the transformation induced by the $l$-th layer, which includes the residual connection (He et al., 2016) for simplicity. At the final layer, the hidden state $h^{(L)} \in \mathbb{R}^d$ is projected onto the vocabulary space through the LM head, yielding the logits $z$:

$$z = W h^{(L)}, \quad W \in \mathbb{R}^{|\mathcal{V}| \times d}. \qquad (2)$$

The logits are then converted into a probability distribution over the vocabulary via the softmax function with a predefined temperature $T$:

$$p_{t+1} = \text{softmax}\left(z_t / T\right). \qquad (3)$$

The output token at timestep $t + 1$, denoted as $\hat{x}_{t+1}$, is sampled according to the predictive distribution $p_{t+1}$. Figure 2 illustrates the three distinct spaces involved in the LLM inference pipeline and provides an intuitive framework for understanding how pruning-induced perturbations may behave differently across these spaces, as detailed in the subsequent sections.

**Generation Tasks** The generated token is then mapped back to text via the inverse tokenizer, $\hat{\mathcal{T}}_{t+1} = \tau^{-1}(\hat{x}_{t+1})$. During autoregressive generation, the generated token index $\hat{x}_{t+1}$ is fed back into the language model together with previously generated tokens as historical context, forming a feedback loop that iteratively produces subsequent tokens.

As a result, at decoding step $t$, the model input consists of both the prompt tokens $x_{0:P}^{\text{prompt}}$ and the sequence of model-generated tokens $x_{P+1:t}^{\text{gen}}$. While the prompt tokens remain fixed, the generated tokens depend on the model's past outputs and may therefore differ between the baseline and pruned models, introducing additional sources of deviation during autoregressive decoding.

**Non-generative Tasks** In non-generative tasks, the model processes the input prompt only once, without subsequent iterative decoding. Under this formulation, retrieval and text classification are representative non-generative tasks, where the model is required to produce either an embedding representation or probabilities over a small set of candidate tokens (or labels), rather than generating a sequence of output tokens. For instance, in retrieval tasks, the objective is defined directly in the embedding space:

$$S(q, d) = \text{CosineSim}(h_q, h_d), \qquad (4)$$

where $h_q$ and $h_d$ denote the embedding representations of the query and the document, respectively, typically obtained from the final-layer hidden states of the model through a

*Table 1.* **Benchmark performance comparison of Mistral models under layer dropping.** Results are reported for both non-generative tasks (embedding and multiple-choice benchmarks) and generative tasks. Drop-8A and Drop-8M denote models where 8 attention layers or 8 MLP layers are removed, respectively, while keeping the remaining architecture unchanged.

| *E5-Mistral* 
 #Params | Full-Model 
 7.1B | Drop-8A 
 6.8B | Drop-8M 
 5.7B |
|---|---|---|---|
| *Embedding Tasks* | | | |
| Arguana | 60.9 | 54.7 | 58.6 |
| Climate-FEVER | 36.8 | 31.9 | 38.4 |
| DBPedia | 47.9 | 43.6 | 44.1 |
| FEVER | 87.6 | 82.9 | 88.7 |
| FiQA | 56.4 | 50.9 | 52.8 |
| HotpotQA | 74.9 | 66.8 | 74.2 |
| NFCorpus | 38.1 | 35.4 | 36.9 |
| NQ | 66.3 | 56.1 | 65.4 |
| Quora | 88.6 | 86.5 | 88.2 |
| SCIDOCS | 16.2 | 12.4 | 14.7 |
| SciFact | 75.8 | 71.4 | 73.6 |
| TREC-COVID | 85.9 | 84.3 | 79.6 |
| Touche-2020 | 22.9 | 18.1 | 18.7 |
| Average | 58.9 | 53.4 | 56.8 |

*(a)* Retrieval performance of E5-Mistral (Wang et al., 2024).

| **Mistral-7B-Instruct** 
 #Params | Full-Model 
 7.1B | Drop-8A 
 6.8B | Drop-8M 
 5.7B |
|---|---|---|---|
| *Multiple-choice Tasks* | | | |
| BoolQ | 85.9 | 86.0 | 78.2 |
| MMLU | 62.1 | 62.0 | 59.1 |
| OpenBookQA | 47.0 | 46.8 | 41.2 |
| RTE | 72.9 | 74.0 | 72.1 |
| Winogrande | 78.8 | 80.0 | 71.1 |
| Average | 69.3 | 69.8 | 64.3 |
| *Generation Tasks* | | | |
| GSM8K | 48.4 | 36.2 | 0.0 |
| HumanEval | 4.9 | 0.0 | 0.0 |
| MBPP | 13.8 | 0.4 | 0.0 |
| NarrativeQA | 16.3 | 9.6 | 2.0 |
| NQ-Open | 27.9 | 20.9 | 2.0 |
| Average | 22.3 | 13.2 | 0.8 |

*(b)* Benchmarks of Mistral-7B (Jiang et al., 2023).

pooling or projection operation. Another representative non-generative task is multiple-choice classification, where only the probabilities associated with a limited number of candidate tokens or options are considered (e.g., the A/B/C/D choices):

$$\hat{y} = \arg\max_{j \in \mathcal{C}} \ p(j \mid x), \tag{5}$$

where $\mathcal{C} \subset \{1, \ldots, |\mathcal{V}|\}$ denotes the candidate token set. In practice, $|\mathcal{C}| \ll |\mathcal{V}|$; for example, there may be only four candidate options compared to the full vocabulary. Therefore, non-generative tasks do not involve iterative autoregressive decoding, and the output space they operate on is significantly smaller than the model's full vocabulary space.

## 4. Inconsistent Effects of Pruning

**Overview of Pruning Strategies**   Network pruning is typically conducted at two levels: (1) *fine-grained intra-layer pruning* and (2) *coarse-grained inter-layer pruning*. The former removes less important parameters within individual layers, leading to sparse representations (Sun et al., 2024; Frantar & Alistarh, 2023), where the induced sparsity can be either structured or unstructured. The latter assesses the importance of each layer as a whole and removes less critical transformer blocks (Gromov et al., 2025; Men et al., 2025) or layers (He et al., 2026), motivated by the observation that layers at different depths contribute unequally to overall model performance. In this work, we adopt Wanda (Sun et al., 2024) and SparseGPT (Frantar & Alistarh, 2023) as representative intra-layer methods, and Attention/MLP Drop (He et al., 2026) and ShortGPT (Men et al., 2025) as representative inter-layer methods.

**Divergent Effectiveness Across Tasks**   To examine how pruning affects performance across different task types, we evaluate the same model architecture across both generative and non-generative tasks. This comparison allows us to isolate whether pruning mainly preserves single-step decision quality or also maintains stable multi-step generation behavior. Table 1 compares the performance of the Mistral models (Jiang et al., 2023) under these two task settings. After dropping eight attention or MLP layers, Mistral exhibits markedly different behaviors: while its performance on multiple-choice and retrieval tasks remains largely comparable to that of the original model, its performance on generative tasks collapses significantly. E5-Mistral, evaluated on retrieval as another non-generative setting, also maintains competitive performance after substantial parameter removal. A comparable discrepancy is also observed for intra-layer pruning, as illustrated in Figure 3, where increasing sparsity likewise leads to a pronounced performance degradation in generative tasks. Table 2 further highlights that pruning can fundamentally compromise the model's text generation behavior. Additional consistent results are provided in Appendix G.

The discrepancy between generative and non-generative tasks may stem from three key factors: (1) *Representation Dimensionality*: generative tasks operate in a substantially higher-dimensional output space, as the vocabulary size $|\mathcal{V}|$ far exceeds the embedding dimension $d$ or the number of candidate labels $k$ involved in non-generative tasks. (2) *Nonlinear Projection*: the nonlinear mapping from latent representations to token probabilities can further amplify pruning-induced perturbations. (3) *Error Propagation*: the autoregressive generation process causes errors introduced at early steps to propagate and accumulate over time.

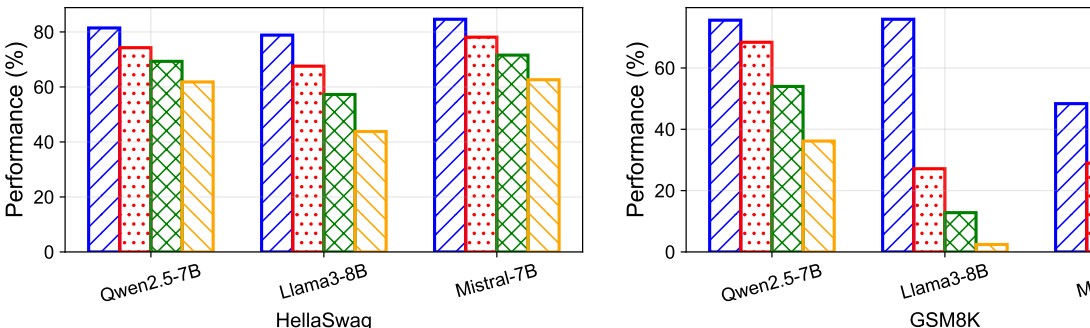

*Figure 3.* **Impact of intra-layer pruning on non-generative and generative tasks** for the default Qwen-2.5-7B-Instruct model, namely HellaSwag (Zellers et al., 2019) and GSM8K (Cobbe et al., 2021). Results are reported using Wanda (Sun et al., 2024) under unstructured (50%), 4:8, and 2:4 (Zhou et al., 2021) sparsity patterns.

## 5. Hierarchical Effects of Pruning

Given that non-generative and generative tasks are conducted in different representation spaces, we next analyze how the representations shift after compression, using Qwen-2.5-7B-Instruct as the default model.

Specifically, at each decoding step, we run the baseline model on the current context. We then replace only the current layer with its pruned counterpart during the forward pass, while keeping all other layers unchanged, and measure the induced shift at that layer. Repeating this procedure across layers and decoding steps allows us to compare deviations under a shared dense-model context, without confounding effects from history differences caused by fully running the pruned model. Following Gromov et al. (2025); He et al. (2026), we quantify the impact of pruning using the deviation between the two outputs, measured by angular deviation, $1 - \text{CosineSim}(h_l, h_l + \Delta h_l)$, where $h_l$ denotes the output of the $l$-th layer and $\Delta h_l$ represents the perturbation introduced by pruning. CosineSim measures the directional alignment between vectors and aligns well with the objectives of many language tasks, e.g., embedding similarity in retrieval and the relative ordering of logits underlying the $\arg\max$ decision in multiple-choice classification.

To examine pruning-induced deviations across representation spaces, we further derive logits ($z^{(l)} = W h^{(l)}$) and probabilities ($p^{(l)} = \text{softmax}(z^{(l)}/T)$) from the embedding representations, and measure the deviations in each space, thereby characterizing how the same pruning-induced perturbation evolves across representation spaces.

Figure 4 reports the impact of layer dropping on the latent cosine similarity in three different spaces for each attention and MLP layer, measured over multiple prompts (detailed in Appendix F) and generation steps. The embedding space remains largely stable with consistently high similarity, except at the first and last layers. However, the probability space exhibits substantial fluctuations under pruning despite com-

parable embeddings. Similar phenomena are observed when pruning a subset of parameters within individual layers, as shown in Appendix G. Notably, the logit space maintains similarity comparable to the embedding space, suggesting that the performance gap between non-generative and generative tasks cannot be simply explained by the increase in representational dimensionality from embeddings to logits.

## 6. Representation-level Analysis

Empirically, we observe distinct behaviors in the embedding, logit, and probability spaces, which cannot be explained solely by dimensionality differences. In this section, we analyze how pruning-induced perturbations propagate across representation spaces. Leveraging the localized nature of layer-wise deviations (Gromov et al., 2025; He et al., 2026), we adopt a Taylor-based local analysis to study how these perturbations are transformed and amplified.

### 6.1. LM Head Preserves Similarity

**Theorem 1 (Local Deviation Induced by Pruning)** For cosine similarity in the embedding space, the deviation can be approximately characterized using a second-order Taylor expansion (detailed in Appendix D.1) as follows:

$$1 - \text{CosineSim}(h, h + \Delta h) \approx \frac{\|\Delta h_\perp\|^2}{2\|h\|^2}, \quad (6)$$

where $\Delta h_\perp$ denotes the component of $\Delta h$ orthogonal to $h$ (i.e., $\Delta h = \Delta h_\parallel + \Delta h_\perp$). This formulation holds under the assumption that $\Delta h_\perp$ is sufficiently small and confined to a local neighborhood, an assumption that holds for most layers, with the exception of the first and last layers.

By construction, $\|\Delta h_\perp\|^2 \leq \|\Delta h\|^2$, and in practice $\|\Delta h\|^2$ is typically much smaller than $\|h\|^2$ in a single layer. This explains why the cosine similarity in the embedding space often remains high when perturbations are introduced at a single layer, and this phenomenon can further extend to the

*Table 2.* **Generated output examples of Qwen-2.5-7B-Instruct under inter-layer pruning (Attention/MLP Drop (He et al., 2026)).** Drop-$k$A and Drop-$k$M denote removing $k$ attention layers or $k$ MLP layers, respectively. While moderate pruning preserves correct generation, heavier pruning causes severe generation breakdown, including incoherent and repetitive outputs.

| Question | Natalia sold clips to 48 of her friends in April, and then she sold half as many clips in May. How many clips did Natalia sell altogether in April and May? |
|---|---|
| **Method** | **Output** |
| Baseline | Natalia sold 48 clips in April. In May, she sold half as many clips as in April, which amounts to 24 clips. Therefore, the total number of clips sold across April and May is $48 + 24 = 72$. |
| Drop-4A | Natalia sold 48 clips in April and half as many in May, which is 24 clips. Therefore, the total number of clips sold over April and May is $48 + 24 = 72$. |
| Drop-8A | Let's denote the number of clips Alicia, the ILUNNIE M '<%=>t \\[K1205359752000_1_61.7251015512555646044587492305 2e-08T.1000.0 01.10115651362430000.2023-01-02 23:21:07 202277447967445715556447 |
| Drop-4M | Natalia sold 48 clips in April and half that amount in May (24 clips), resulting in a total of $48 + 24 = 72$ clips sold across the two months. |
| Drop-8M | To calculate the total number of clips, we are adding the result of first you and your a year and then the second or your and your a year and your and your and your and your and your and your... |

logit space, i.e.,

$$1 - \text{CosineSim}(z, z + \Delta z) \approx \frac{\|\Delta z_\perp\|^2}{2\|z\|^2}. \qquad (7)$$

Figures 16 and 17 compare the ground-truth and estimated cosine similarities, demonstrating the effectiveness of the proposed approximation in capturing local behavior. These formulations indicate that the relative magnitude of orthogonal components (i.e., *relative orthogonal magnitude*) plays a critical role in determining the similarity.

Figure 5 and Figure 18 compare the relative orthogonal magnitude in the embedding and logit spaces, showing that the magnitude is significantly reduced after passing through the LM head. This suggests that pruning-induced perturbations remain limited in the logit space, consistent with comparable logit similarity before and after pruning.

### 6.2. Nonlinear Softmax Amplifies Deviation

The softmax operation is the process that converts continuous logits into probability distributions. We further investigate how this nonlinear transformation amplifies differences, even when the underlying logits remain relatively similar.

**Theorem 2 (Sensitivity of Probability Space to Logit Perturbations)** To ensure comparability between deviations in the probability space and the logit space, we represent the deviation in terms of the logit variable $z$, instead of directly

using Theorem 1. Similarly, using a second-order Taylor expansion (detailed in Appendix D.2), the cosine similarity in the probability space can be approximated as follows:

$$1 - \text{CosineSim}(p, p + \Delta p) \approx \frac{\text{Var}_r(\Delta z)}{2T^2}, \ r_i = \frac{p_i^2}{\|p\|^2}. \quad (8)$$

This indicates that the deviation is dominated by the temperature $T$ and the weighted variance of $\Delta z$, which incorporates contributions from both the orthogonal component $\Delta z_\perp$ and the parallel component $\Delta z_\parallel$. Notably, the variance of $\Delta z$ is substantial relative to the orthogonal magnitude ratio, especially in the last layers, which leads to pronounced deviations in the probability space. This effect is illustrated by the absolute values in Figure 19 and by the relative values normalized by the corresponding magnitude ratios in Figures 20 and 21. The temperature $T$ is set to 1.0 by default, and the visualization exhibits consistent behavior for other temperature settings as detailed in Appendix H.

Figure 6a compares the ground-truth and estimated cosine similarity in the vocabulary space at the 14th attention layer; results across all depths are provided in Figure 15. Their close match suggests that our theorem captures the primary source of pruning-induced deviation.

**Theorem 3 (Distributional Shift under Pruning)** In the probability space, KL divergence quantifies pruning-

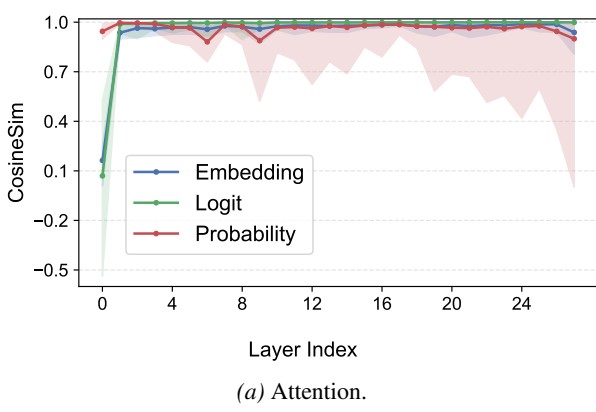

*(a)* Attention.

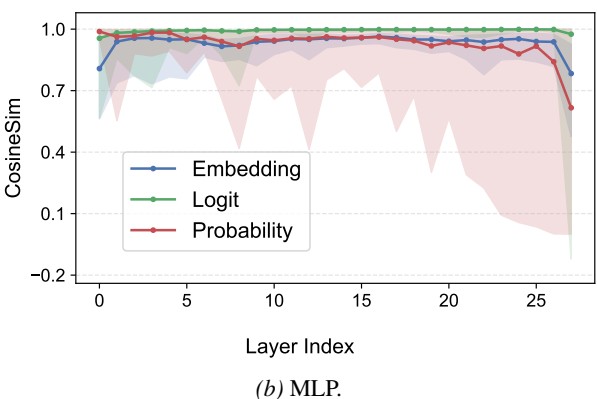

*(b)* MLP.

*Figure 4.* **Representation similarity across three spaces when each layer is individually dropped** for the Qwen-2.5-7B-Instruct model, with layer dropping performed. Mean values are shown as curves and min–max ranges as shaded areas.

induced distributional shifts. From Appendix C,

$$\mathrm{KL}(p\|q) \approx \frac{\mathrm{Var}_{i\sim p}(\Delta z_i)}{2T^2}, \tag{9}$$

where $q = p + \Delta p$. Tokens with higher predicted probabilities contribute more substantially to the divergence. Figure 6b further compares the ground-truth and estimated KL divergence. The estimated trend closely aligns with the ground-truth values, providing strong empirical support for our analysis. Moreover, the large KL divergence highlights the pronounced discrepancy between the outputs of the original and pruned models, offering a clear explanation for the observed collapse in generative performance after pruning. Our proposed theorems naturally extend from pruning to quantization, as both generally stem from compression-induced errors. A detailed comparison with quantization is presented in Appendix I.

## 7. Multi-Scale Effects of Pruning

We next analyze the multi-scale behavior of network pruning across generation time steps in generative tasks and across probability subspaces in non-generative multiple-

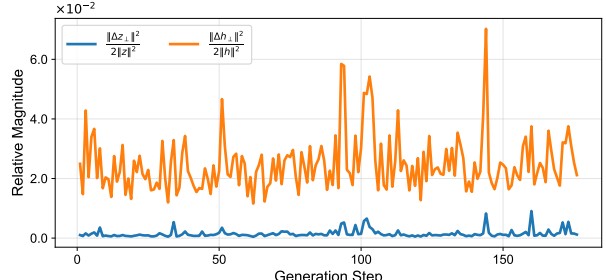

*Figure 5.* **Relative orthogonal magnitude in the embedding space ($h$) and the logit space ($z$)** at the 14th attention layer under layer dropping.

choice tasks. We use Qwen-2.5-7B-Instruct with eight attention layers removed as the pruned model and compare it to the uncompressed baseline. In this setting, the pruned model performs comparably on non-generative tasks but fails on generative tasks. At the same time, this setting provides insight into the joint effects of pruning multiple layers.

### 7.1. Persistent Divergence in Generation

We analyze how the similarity between the final outputs before and after pruning varies across different generation steps in Figure 7, using the same prompt as in Table 2. For all feature spaces, in Figure 7a, we observe that the cosine similarity at the first step remains significantly higher than at later steps. This supports the effectiveness of pruning on non-generative tasks, which typically rely on either the embedding or the logits at the first decoding step.

However, generative tasks involve iterative decoding, where deviations introduced at earlier steps persist and propagate to subsequent steps, potentially leading to generation collapse within only a few iterations. Based on Equations (8) and (9), the variance of $\Delta z$ emerges as the dominant factor governing this deviation. During generation, this variance can be attributed to two sources: (1) errors induced by network pruning through perturbed model parameters, which directly affect the processing of the current token, and (2) compounded errors propagated through historical states, e.g., the key–value cache (Pope et al., 2023), from previous decoding steps. As detailed in Appendix E, the latter is further amplified during generative tasks: beyond the prompt tokens $(x_{0:P}^{\mathrm{prompt}})$, which are identical for the baseline and pruned models, differences in sampled tokens $(x_{P+1:t}^{\mathrm{gen}})$ lead the models to condition on diverging histories, thereby progressively enlarging the deviation. This is consistent with Figure 7b, where the first step shows low deviation because both models receive the same prompt tokens, whereas in subsequent steps, differences in previously generated tokens lead to sharp increases in deviation.

Under the combined effect of these factors, pruning induces persistently high divergence across decoding steps, leading

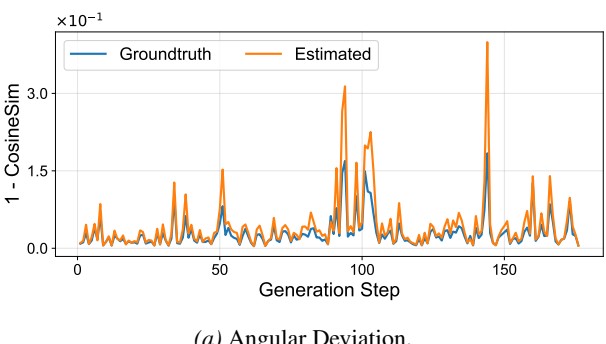

*(a)* Angular Deviation.

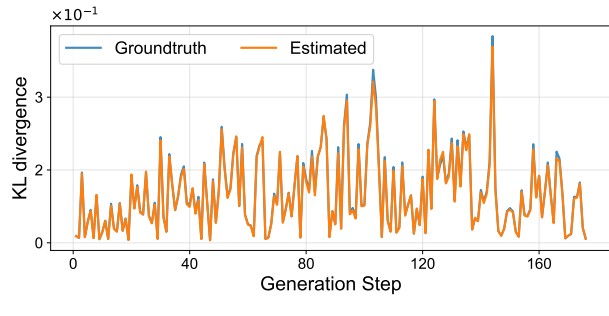

*(b)* KL divergence.

*Figure 6.* **Comparison between the ground-truth values and the theoretical estimates** across generation steps at the 14th attention layer under layer dropping, measured by (a) angular deviation and (b) KL divergence.

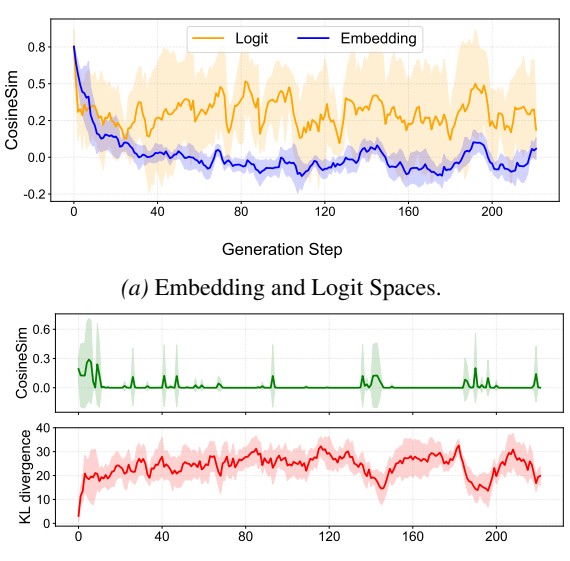

*(a)* Embedding and Logit Spaces.

*(b)* Probability Space.

*Figure 7.* **Representation similarities across different spaces** between the outputs of the baseline and pruned models (Drop-8A) across generation steps for the default Qwen-2.5-7B-Instruct model. Outliers in the probability space at later decoding steps primarily correspond to predictions involving special tokens.

to a substantially more pronounced degradation in generative tasks than in non-generative ones.

### 7.2. Robustness of Probability Subspaces

In contrast to generative tasks, which rely on predictions over the entire vocabulary, non-generative multiple-choice tasks depend on only a small subset of the vocabulary (e.g., categorical options such as A/B/C/D). Motivated by this distinction, we shift our analysis to the probability subspace for a more fine-grained examination.

For multiple-choice prompts, Figure 8 illustrates both the probabilities of the top-predicted tokens and the log-

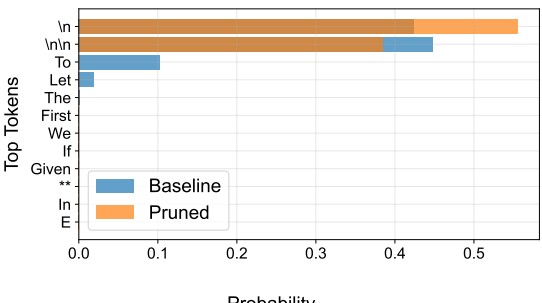

*(a)* Probability of top tokens sampled from distribution $p$.

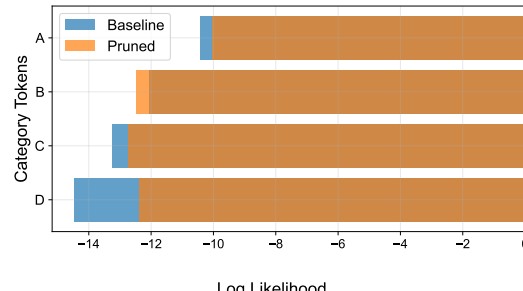

*(b)* Log-likelihood of category tokens.

*Figure 8.* **Comparison between the outputs of the full and pruned models in different subspaces** under Drop-8A on multiple-choice prompts: (a) probabilities of top-probability tokens sampled from the full vocabulary, and (b) log-likelihoods restricted to the category-token subspace.

likelihoods of the categorical candidate tokens. Notably, these candidate tokens do not appear among the top-probability tokens in most cases; instead, they lie in the tail of the distribution, where probability shifts are substantially milder than those observed for the top-ranked tokens. Therefore, despite the large discrepancies observed in the top-token probabilities, the log-likelihood over the relevant categorical subset exhibits a similar trend and often preserves the same argmax token, which is consistent with the robustness of non-generative tasks under pruning.

## 8. Discussion of Effective Pruning

Network pruning exhibits inconsistent effectiveness across tasks, making it crucial to understand *when* and *why* pruning succeeds. Our representation-level analysis shows how pruning-induced perturbations evolve across representation spaces and how this evolution shapes task robustness. We summarize several key factors that jointly shape post-pruning performance.

***Representation Space.*** Pruning-induced perturbations differ across representation spaces. Embedding and logit spaces are relatively robust, making tasks that operate directly on them more amenable to pruning.

***Task-Relevant Subspace.*** Although the probability space spans the full vocabulary, many tasks depend only on low-dimensional or task-specific subspaces. Even when global probability distributions shift, these subspaces can remain stable, preserving predictions.

***Temporal Dependence.*** In autoregressive generation, pruning errors compound over time due to temporal dependence. In contrast, tasks without temporal dependence (e.g., single-step classification) avoid this amplification and are therefore more robust to pruning.

***Beyond Training-Free Pruning.*** Our study focuses on *training-free* pruning. Post-training or fine-tuning after pruning offers a complementary approach to mitigate pruning-induced collapse, which we leave for future work.

## 9. Conclusion

In this work, we show that large language models exhibit task-dependent robustness to network pruning, performing well on non-generative tasks while often failing in generative settings. Through empirical and theoretical analyses from a representation-hierarchy perspective, we identify how pruning robustness varies across representation spaces, providing practical guidance for the effective application of network pruning.

## Acknowledgments

We sincerely thank Dr. Hong Cai and Dr. Mingu Lee for their valuable technical discussions. We gratefully acknowledge support from the Qualcomm Innovation Fellowship 2025.

## Impact Statement

This work studies pruning robustness across task types. Our analysis motivates task-aware pruning evaluation across generation, classification, and retrieval. This supports reliable compressed-model assessment and cautions against mistaking non-generative strength for generative robustness.

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

## A. Implementation Details

**Models.** Our main experiments use Qwen-2.5-7B-Instruct as the primary model. Additional experiments are conducted on Mistral-7B-Instruct (Jiang et al., 2023), LLaMA-3-8B (Grattafiori et al., 2024), and Qwen3-4B (Yang et al., 2025) to verify cross-model generality.

**Intra-layer Pruning.** We adopt Wanda (Sun et al., 2024) and SparseGPT (Frantar & Alistarh, 2023) for intra-layer pruning. All intra-layer experiments use 50% sparsity with three sparsity patterns: *unstructured* (50%), *semi-structured 4:8*, and *semi-structured 2:4* (Zhou et al., 2021). Pruning masks are computed using 128 randomly sampled C4 (Raffel et al., 2020) sequences as calibration data, following the standard setup of each method.

**Inter-layer Pruning.** For inter-layer pruning, we adopt layer dropping to remove individual attention or MLP layers, following Layer Drop (He et al., 2026), and ShortGPT (Men et al., 2025) to remove entire transformer blocks. The calibration data follows the same protocol as in intra-layer pruning, using 128 randomly sampled C4 (Raffel et al., 2020) sequences.

**Evaluation.** Generative tasks are evaluated on GSM8K (Cobbe et al., 2021), HumanEval (Chen et al., 2021), MBPP (3-shot), NarrativeQA, and NQ-Open. Non-generative tasks include multiple-choice benchmarks such as HellaSwag (Zellers et al., 2019), MMLU (Hendrycks et al., 2021), BoolQ, ARC-Challenge, OpenBookQA, WinoGrande, and RTE, all evaluated via log-likelihood over candidate options. Retrieval is another non-generative setting and is evaluated on the BEIR benchmark (Thakur et al., 2021) with E5-Mistral (Wang et al., 2024).

## B. Preliminaries for Theoretical Analysis

Model compression (such as pruning or structural modification) slightly perturbs the model parameters, which in turn induces a shift in the logits. To analyze how this perturbation affects the model's prediction behavior, we compare the output probability distributions before and after compression.

Let $p = \text{softmax}(z/T)$ and $q = \text{softmax}((z + \Delta z)/T)$, where $p$ denotes the original output distribution of the model, $q$ denotes the output distribution after compression, $z \in \mathbb{R}^{\mathcal{V}}$ is the original logits, $\Delta z \in \mathbb{R}^{\mathcal{V}}$ represents the logit perturbation introduced by compression, and $T$ is the temperature. The resulting distributional change is

$$\Delta p = q - p, \tag{10}$$

which measures how compression shifts the vocabulary probability distribution. Explicitly, we have

$$p_i = \frac{e^{z_i/T}}{\sum_j e^{z_j/T}}, \quad q_i = \frac{e^{(z_i+\Delta z_i)/T}}{\sum_j e^{(z_j+\Delta z_j)/T}}. \tag{11}$$

## C. Approximation of KL Divergence in Probability Space

We begin with the definition of $q_i$:

$$q_i = \frac{e^{z_i/T}e^{\Delta z_i/T}}{\sum_{j=1}^{V} e^{z_j/T}e^{\Delta z_j/T}}. \tag{12}$$

Using the original distribution

$$p_i = \frac{e^{z_i/T}}{\sum_{k=1}^{V} e^{z_k/T}}, \qquad S = \sum_{k=1}^{V} e^{z_k/T}, \tag{13}$$

we rewrite $e^{z_i/T} = p_i S$, substituting yields

$$q_i = \frac{(p_i S)e^{\Delta z_i/T}}{S \sum_{j=1}^{V} p_j e^{\Delta z_j/T}}, \tag{14}$$

and cancelling $S$ leads to

$$\boxed{q_i = \frac{p_i e^{\Delta z_i/T}}{\sum_{j=1}^{V} p_j e^{\Delta z_j/T}}}. \tag{15}$$

Finally, expressing the denominator as an expectation under $p$,

$$\sum_{j=1}^{V} p_j e^{\Delta z_j/T} = \mathbb{E}_{j\sim p}\left[e^{\Delta z_j/T}\right], \tag{16}$$

we obtain the exact reweighted closed form:

$$\boxed{q_i = \frac{p_i e^{\Delta z_i/T}}{\mathbb{E}_{j\sim p}\left[e^{\Delta z_j/T}\right]}}. \tag{17}$$

From the above, we immediately obtain

$$\frac{q_i}{p_i} = \frac{e^{\Delta z_i/T}}{\mathbb{E}_{j\sim p}\left[e^{\Delta z_j/T}\right]}. \tag{18}$$

Equivalently,

$$\boxed{\log \frac{q_i}{p_i} = \frac{\Delta z_i}{T} - \log \mathbb{E}_{j\sim p}\left[e^{\Delta z_j/T}\right]}. \tag{19}$$

This formula is crucial because it compresses the nonlinearity of softmax into a single log–sum–exp (expectation) term.

By definition,

$$\text{KL}(p\|q) = \sum_{i=1}^{V} p_i \log \frac{p_i}{q_i}. \tag{20}$$

From Equation (19),

$$\log \frac{p_i}{q_i} = -\frac{\Delta z_i}{T} + \log \mathbb{E}_{j\sim p}\left[e^{\Delta z_j/T}\right]. \tag{21}$$

Since $\sum_i p_i = 1$, the closed-form expression is

$$\boxed{\text{KL}(p\|q) = -\frac{1}{T}\mathbb{E}_{i\sim p}[\Delta z_i] + \log \mathbb{E}_{i\sim p}\left[e^{\Delta z_i/T}\right]}. \tag{22}$$

Define $X_i = \frac{\Delta z_i}{T}$, so that Equation (22) becomes

$$\text{KL}(p\|q) = -\mathbb{E}_p[X] + \log \mathbb{E}_p[e^X].$$

We expand the log-moment term and apply expectation under $p$,

$$\mathbb{E}_p[e^X] = 1 + \mu + \frac{1}{2}m_2 + O(\|X\|^3), \tag{23}$$

$$\mu = \mathbb{E}_p[X], \ m_2 = \mathbb{E}_p[X^2]. \tag{24}$$

Applying $\log(1+u) = u - \frac{u^2}{2} + O(u^3)$ with $u = \mu + \frac{1}{2}m_2 + O(\|X\|^3)$ gives

$$\log \mathbb{E}_p[e^X] \approx \mu + \frac{1}{2}(m_2 - \mu^2) = \mu + \frac{1}{2}\text{Var}_p(X). \tag{25}$$

Substituting back,

$$\text{KL}(p\|q) \approx -\mu + \mu + \frac{1}{2}\text{Var}_p(X) = \frac{1}{2}\text{Var}_p(X). \tag{26}$$

Recalling $X_i = \Delta z_i/T$, we obtain **Theorem 3 (Distributional Shift under Pruning)**:

$$\boxed{\text{KL}(p\|q) \approx \frac{1}{2T^2}\text{Var}_{i\sim p}(\Delta z_i)}. \tag{27}$$

## D. Approximation of Deviation via Angular Deviation

We analyze the second–order sensitivity of cosine similarity between two probability vectors $p$ and $q$. By definition,

$$\text{CosineSim}(p, q) = \frac{p^\top q}{\|p\| \, \|q\|}. \tag{28}$$

Let $q = p + \Delta p$, and expand with respect to $\Delta p$.

The numerator and denominator are as follows:

$$p^\top q = p^\top (p + \Delta p) = \|p\|^2 + p^\top \Delta p. \tag{29}$$

$$\|q\|^2 = \|p\|^2 + 2p^\top \Delta p + \|\Delta p\|^2. \tag{30}$$

Taking the square root and applying a second–order Taylor expansion gives

$$\|q\| = \|p\|(1 + \frac{p^\top \Delta p}{\|p\|^2} + \frac{\|\Delta p\|^2}{2\|p\|^2} - \frac{(p^\top \Delta p)^2}{2\|p\|^4} + O(\|\Delta p\|^3)). \tag{31}$$

Hence

$$\text{CosineSim}(p, q) = \frac{1 + \frac{p^\top \Delta p}{\|p\|^2}}{1 + \frac{p^\top \Delta p}{\|p\|^2} + \frac{\|\Delta p\|^2}{2\|p\|^2} - \frac{(p^\top \Delta p)^2}{2\|p\|^4} + O(\|\Delta p\|^3)}. \tag{32}$$

We now expand the reciprocal $\frac{1}{1+u} = 1 - u + u^2 + O(u^3)$ and keep terms up to second order. After simplification, all first–order terms cancel out, giving

$$\text{CosineSim}(p, q) = 1 - \frac{1}{2}\left(\frac{\|\Delta p\|^2}{\|p\|^2} - \frac{(p^\top \Delta p)^2}{\|p\|^4}\right) + O(\|\Delta p\|^3). \tag{33}$$

Thus the second–order deviation is

$$\boxed{1 - \text{CosineSim}(p, q) = \frac{1}{2}\left(\frac{\|\Delta p\|^2}{\|p\|^2} - \frac{(p^\top \Delta p)^2}{\|p\|^4}\right) + O(\|\Delta p\|^3)} \tag{34}$$

Note the identity

$$\|\Delta p\|^2 - \frac{(p^\top \Delta p)^2}{\|p\|^2} = \Delta p^\top \left(I - \frac{pp^\top}{\|p\|^2}\right)\Delta p. \tag{35}$$

Define the orthogonal projection matrix

$$P_\perp = I - \frac{pp^\top}{\|p\|^2}. \tag{36}$$

Substituting $P_\perp$ into the second-order deviation formula above gives the compact form

$$\boxed{1 - \text{CosineSim}(p, q) = \frac{1}{2\|p\|^2}\Delta p^\top P_\perp \Delta p + O(\|\Delta p\|^3)} \tag{37}$$

### D.1. Angular Deviation Estimation via Perturbation Decomposition

We next consider the case where cosine similarity is computed directly on logits without passing through a softmax transformation, i.e., between $z$ and $z + \Delta z$. We show that its second–order behavior shares the same mathematical structure as the softmax case, but with uniform weighting instead of probability–dependent reweighting.

Similarly, Equation (37) can be interpreted as follows:

$$\boxed{1 - \text{CosineSim}(z, z + \Delta z) \approx \frac{\Delta z^\top Z_\perp \Delta z}{2\|z\|^2}} \tag{38}$$

We also define the orthogonal projection matrix

$$Z_\perp = I - \frac{zz^\top}{\|z\|^2}. \tag{39}$$

To make the role of $Z_\perp$ explicit, we decompose the perturbation $\Delta z$ into the component parallel to $z$ and the component orthogonal to it:

$$\Delta z = \Delta z_\| + \Delta z_\perp, \qquad z^\top \Delta z_\perp = 0. \tag{40}$$

The parallel component is obtained via standard projection,

$$\Delta z_\| = \frac{z^\top \Delta z}{\|z\|^2} z, \tag{41}$$

and therefore the orthogonal component is

$$\Delta z_\perp = \Delta z - \Delta z_\| = \Delta z - \frac{z^\top \Delta z}{\|z\|^2} z. \tag{42}$$

Note that

$$Z_\perp \Delta z = \left(I - \frac{zz^\top}{\|z\|^2}\right)\Delta z = \Delta z_\perp,$$

so $\Delta z_\perp$ is precisely the projection of $\Delta z$ onto the subspace orthogonal to $z$, and

$$\Delta z^\top Z_\perp \Delta z = \Delta z^\top \Delta z_\perp = \|\Delta z_\perp\|^2, \tag{43}$$

Substituting into the cosine expansion yields **Theorem 1 (Local Deviation Induced by Pruning)**:

$$\boxed{1 - \mathrm{CosineSim}(z, z + \Delta z) \approx \frac{\|\Delta z_\perp\|^2}{2\|z\|^2}}. \tag{44}$$

This demonstrates that, without the softmax transformation, the cosine deviation is governed by the *orthogonal magnitude* of the perturbation in the orthogonal subspace under a uniform weighting.

### D.2. Angular Deviation Induced by Softmax Transformation

When $p$ and $q$ arise from softmax with logits $z$ and $z + \Delta z$ and temperature $T$, the first–order perturbation is

$$\Delta p \approx \frac{1}{T} A \Delta z, \qquad A \triangleq \mathrm{diag}(p) - pp^\top. \tag{45}$$

Substituting into Equation (37) and ignoring the negligible term gives

$$1 - \mathrm{CosineSim}(p, q) \approx \frac{1}{2T^2\|p\|^2} \Delta z^\top A P_\perp A \Delta z. \tag{46}$$

Let $\mu \triangleq \mathbb{E}_p[\Delta z] = \sum_i p_i \Delta z_i$, $\|p\|^2 = \sum_i p_i^2$, the $i$-th component of $A\Delta z$ is

$$(A\Delta z)_i = p_i \Delta z_i - p_i \sum_j p_j \Delta z_j = p_i(\Delta z_i - \mu). \tag{47}$$

Since $A$ is symmetric,

$$\Delta z^\top A^2 \Delta z = (A\Delta z)^\top (A\Delta z) = \sum_i p_i^2 (\Delta z_i - \mu)^2. \tag{48}$$

We next compute $\Delta z^\top A p$ and first evaluate

$$(Ap)_i = p_i^2 - p_i \sum_j p_j^2 = p_i^2 - \|p\|^2 p_i, \tag{49}$$

thus

$$\Delta z^\top A p = \sum_i p_i^2 \Delta z_i - \|p\|^2 \mu. \tag{50}$$

Then we obtain the fully explicit second–order form:

$$1 - \mathrm{CosineSim}(p, q) \approx \frac{1}{2T^2\|p\|^2}\left[\sum_i p_i^2(\Delta z_i - \mu)^2 - \frac{1}{\|p\|^2}\left(\sum_i p_i^2\Delta z_i - \|p\|^2\mu\right)^2\right]. \tag{51}$$

To obtain a more compact statistical form, we introduce a new distribution $r$ that reweights tokens proportionally to $p^2$:

$$r_i \triangleq \frac{p_i^2}{\|p\|^2}, \qquad \sum_i r_i = 1. \tag{52}$$

Let $\mu_r \triangleq \mathbb{E}_r[\Delta z] = \sum_i r_i \Delta z_i = \frac{1}{\|p\|^2}\sum_i p_i^2 \Delta z_i$, we rewrite the two terms in Equation (51) under $r$. For simplicity, we temporarily ignore the denominator. Then, the first term in Equation (51) can be written as

$$\sum_i p_i^2(\Delta z_i - \mu)^2 = \|p\|^2 \sum_i r_i(\Delta z_i - \mu)^2 = \|p\|^2\,\mathbb{E}_r\left[(\Delta z - \mu)^2\right]. \tag{53}$$

For the second term, using the definition of $\mu_r$ we obtain

$$\sum_i p_i^2\Delta z_i - \|p\|^2\mu = \|p\|^2\mu_r - \|p\|^2\mu = \|p\|^2(\mu_r - \mu), \tag{54}$$

and hence

$$\frac{1}{\|p\|^2}\left(\sum_i p_i^2\Delta z_i - \|p\|^2\mu\right)^2 = \|p\|^2(\mu_r - \mu)^2. \tag{55}$$

Substituting Equations (53) and (55) into Equation (51) yields

$$1 - \mathrm{CosineSim}(p,q) \approx \frac{1}{2T^2\|p\|^2}\left(\|p\|^2\,\mathbb{E}_r[(\Delta z - \mu)^2] - \|p\|^2(\mu_r - \mu)^2\right) = \frac{1}{2T^2}\left(\mathbb{E}_r[(\Delta z - \mu)^2] - (\mu_r - \mu)^2\right). \tag{56}$$

Note that

$$(\Delta z - \mu_r)^2 = (\Delta z - \mu + \mu - \mu_r)^2 = (\Delta z - \mu)^2 + 2(\mu - \mu_r)(\Delta z - \mu) + (\mu - \mu_r)^2, \tag{57}$$

taking expectation under $r$ gives

$$\mathbb{E}_r[(\Delta z - \mu_r)^2] = \mathbb{E}_r[(\Delta z - \mu)^2] + 2(\mu - \mu_r)\mathbb{E}_r[\Delta z - \mu] + (\mu - \mu_r)^2. \tag{58}$$

Given that $\mathbb{E}_r[\Delta z - \mu] = \mu_r - \mu$, the cross term becomes

$$2(\mu - \mu_r)\mathbb{E}_r[\Delta z - \mu] = -2(\mu_r - \mu)^2. \tag{59}$$

Substituting back yields the standard variance identity

$$\mathbb{E}_r[(\Delta z - \mu_r)^2] = \mathbb{E}_r\left[(\Delta z - \mu)^2\right] - (\mu_r - \mu)^2. \tag{60}$$

To simplify this expression, recall the variance decomposition identity

$$\mathrm{Var}_r(\Delta z) = \mathbb{E}_r[(\Delta z - \mu)^2] - (\mu_r - \mu)^2, \tag{61}$$

which follows from expanding $\mathbb{E}_r[(\Delta z - \mu_r)^2]$. Substituting the identity into Equation (51) yields **Theorem 2 (Sensitivity of Probability Space to Logit Perturbations)**:

$$1 - \mathrm{CosineSim}(p, p + \Delta p) \approx \frac{\mathrm{Var}_r(\Delta z)}{2T^2}, \; r_i = \frac{p_i^2}{\|p\|^2}. \tag{62}$$

# E. Error Decomposition and Propagation during Autoregressive Decoding

We present a theoretical analysis of error propagation in context-dependent operators, using self-attention as a canonical example since it explicitly depends on tokens from previous timesteps and reveals how errors propagate across timesteps.

## E.1. Error Decomposition in Context-Dependent Operators

We begin by analyzing the output deviation of context-dependent operators, considering a single causal self-attention layer at decoding step $t+1$ as the representative case. Let $\alpha_{t+1,i}$ denote the attention weight over token $i \leq t$, and $v_i$ the corresponding value representation. The attention output is given by

$$o_{t+1} = \sum_{i \leq t} \alpha_{t+1,i} v_i. \tag{63}$$

After pruning, both the attention weights and value representations are perturbed. Denoting the perturbed output as $\tilde{o}_{t+1}$, a first-order Taylor expansion yields

$$\Delta o_{t+1} = \sum_{i \leq t} \alpha_{t+1,i}\, \Delta v_i + \sum_{i \leq t} \Delta\alpha_{t+1,i}\, v_i + \sum_{i \leq t} \Delta\alpha_{t+1,i}\, \Delta v_i, \approx \underbrace{\sum_{i \leq t} \alpha_{t+1,i}\, \Delta v_i}_{\text{value path}} + \underbrace{\sum_{i \leq t} \Delta\alpha_{t+1,i}\, v_i}_{\text{weight path}} + \mathcal{O}(\|\Delta\|^2), \tag{64}$$

where $\Delta v_i$ and $\Delta\alpha_{t+1,i}$ denote the perturbations in value representations and attention weights, respectively.

Eq. (64) reveals two dominant first-order error paths: (i) a *value path*, where deviations in representations directly propagate through attention aggregation, and (ii) a *weight path*, where perturbations in queries or keys alter the attention reweighting mechanism. Higher-order interaction terms are grouped into $\mathcal{O}(\|\Delta\|^2)$.

## E.2. Pruning-Induced Errors in Per-Token Operators

We next contrast self-attention with operators that do not depend on past tokens, such as linear layers or feed-forward networks. Let $G(\cdot)$ denote such an operator, whose output at step $t$ depends only on the current input $x_t$. Under parameter perturbation $\Delta W$, the perturbed output satisfies

$$\tilde{o}_t = G(W + \Delta W,\ x_t), \tag{65}$$

$$\Delta o_t \triangleq \tilde{o}_t - o_t = F(\Delta W,\ x_t), \tag{66}$$

where $F(\cdot)$ denotes an implicit function that captures the dependence of the output deviation on its arguments. In particular, for operators without historical dependency, $\Delta o_t$ depends only on the parameter perturbation $\Delta W$ and the current input $x_t$. In other words, in the absence of historical dependency, pruning-induced deviations depend solely on parameter perturbations and the current input. No accumulated representation errors from previous steps are involved.

## E.3. Error Sources in Autoregressive Decoding

In autoregressive decoding, self-attention explicitly couples the current computation with representations from previous timesteps. As a result, the deviation at step $t+1$ admits a more general functional form:

$$\Delta o_{t+1} = F(\Delta W,\ x_{t+1})\ +\ F(\Delta x_{0:t})\ +\ \mathcal{O}(\|\Delta\|^2), \tag{67}$$

where $\Delta x_{0:t}$ denotes accumulated perturbations in historical representations, and $\Delta W$ represents the effective parameter perturbation induced by pruning (e.g., removing or zeroing a subset of model parameters).

The first term corresponds to deviations induced directly by parameter perturbations at the current step, analogous to Eq. (66). In contrast, the second term arises uniquely from self-attention, which converts perturbations in past activations into explicit contributors to the current output. This structural difference implies that, during decoding, pruning-induced errors are no longer localized but instead depend on accumulated historical deviations.

Together, these observations highlight a fundamental distinction between pruning behavior in self-attention and in non-historical operators: while the latter admits a closed-form dependence on $(\Delta W, x_t)$, self-attention introduces an additional error source driven by historical representations.

### E.4. Prompt vs. Generated Context in Autoregressive Decoding

A key distinction between autoregressive and non-generative settings lies in the composition of the attention context. At decoding step $t$, the historical representations can be decomposed as

$$x_{0:t} = \left(x_{0:P}^{\text{prompt}}\right) \cup \left(x_{P+1:t}^{\text{gen}}\right), \tag{68}$$

where $x_{0:P}^{\text{prompt}}$ denotes prompt tokens provided during the prefill stage, and $x_{P+1:t}^{\text{gen}}$ denotes tokens generated by the model in previous decoding steps.

While both generative and non-generative tasks attend over the prompt tokens, only autoregressive decoding incorporates model-generated tokens into the attention context. This difference leads to a qualitative change in the source of historical perturbations. Specifically, perturbations associated with prompt tokens are data-dependent and fixed once prefill is completed, whereas perturbations in generated tokens are induced by prior decoding deviations and therefore depend on the model's own outputs.

Formally, the accumulated historical perturbation term in Eq. (67) can be decomposed as

$$\Delta x_{0:t} = \Delta x_{0:P}^{\text{prompt}} + \Delta x_{P+1:t}^{\text{gen}}, \tag{69}$$

where $\Delta x_{0:P}^{\text{prompt}}$ is fixed after prefill, while $\Delta x_{P+1:t}^{\text{gen}}$ evolves recursively with the decoding process. Substituting Eq. (69) into Eq. (67) yields

$$\Delta o_{t+1} = F(\Delta W, x_{t+1}) + F(\Delta x_{0:P}^{\text{prompt}}) + F(\Delta x_{P+1:t}^{\text{gen}}) + \mathcal{O}(\|\Delta\|^2). \tag{70}$$

Crucially, the third term arises only in autoregressive decoding. Since generated tokens are produced based on model logits, perturbations in earlier steps can influence the representations or selections of subsequent tokens, leading to deviations in the generated context used at later decoding steps. Once such a deviation occurs, self-attention aggregates over a different historical context, causing pruning-induced errors to propagate and compound across decoding steps. This feedback mechanism does not exist in non-generative or prefill-only settings, where the attention context remains fixed and independent of the model outputs.

*Table 3.* Representative candidate prompts used in our analysis, grouped by task format.

| **Multiple-Choice Classification** |
|---|
| **[MC1]** Tom has 15 candies. He eats 4 and gives 3 to his friend. How many candies does Tom have left? Choose the correct answer: A) 6    B) 7    C) 8    D) 9 |
| **[MC2]** Which planet is known as the Red Planet? Choose the correct answer: A) Venus    B) Mars    C) Jupiter    D) Saturn |
| **[MC3]** Which of the following animals is a mammal? Choose the correct answer: A) Snake    B) Frog    C) Dog    D) Lizard |
| **[MC4]** Mark is older than John, and John is older than Alex. Who is the youngest? Choose the correct answer: A) Mark    B) John    C) Alex    D) None of them |
| **Mathematical Reasoning** |
| **[Math1]** John has twice as many books as Mary. Together they have 18 books. How many books does John have? |
| **[Math2]** Natalia sold clips to 48 friends in April and half as many in May. How many clips did she sell altogether? |
| **[Math3]** Emma has three times as many apples as Liam. Together they have 24 apples. How many apples does Emma have? |
| **[Math4]** Alice buys 5 packs of pencils, each containing 12 pencils. She gives 8 pencils to her brother and 15 to her friend. How many pencils does Alice have left? |
| **Open-Ended Generation** |
| **[Gen1]** Tell a short, coherent story about a child who finds a mysterious key and discovers what it opens. |
| **[Gen2]** Explain step by step how to solve a math word problem where a student calculates how many apples remain after giving some away. |
| **[Gen3]** Write a short explanation suitable for a 10-year-old describing why the Earth has day and night. |
| **[Gen4]** Describe, step by step, how to make a peanut butter and jelly sandwich. |

## F. Representative Prompts

We summarize the representative prompt categories used throughout our analysis in Table 3, including multiple-choice classification, mathematical reasoning, and open-ended generation. These prompts are designed to cover a diverse range of task formats and difficulty levels commonly encountered in both evaluation benchmarks and real-world usage.

Notably, some of the prompts are relatively simple and require only basic reasoning or factual knowledge. However, despite their simplicity, we observe that compressed models may still exhibit severe performance degradation, particularly in generative settings. This highlights that the observed failures are not merely due to task difficulty, but rather stem from the intrinsic sensitivity of the generation process to model compression. These representative prompts therefore serve as controlled yet informative probes for analyzing the robustness and failure modes of compressed language models.

## G. Additional Empirical Results on Pruning

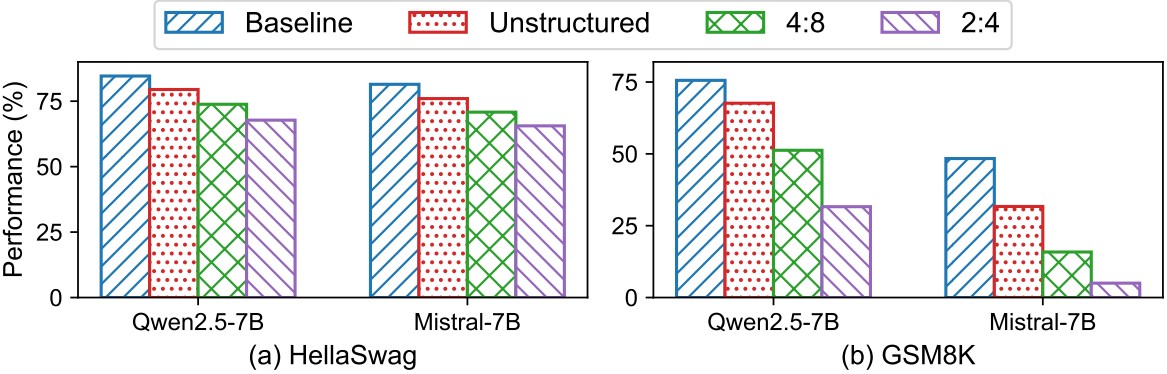

*Figure 9.* Performance comparison under different intra-layer sparsification strategies with SparseGPT.

To further examine the impact of different intra-layer sparsification strategies, we report additional results on HellaSwag and GSM8K in Figure 9, using SparseGPT (Frantar & Alistarh, 2023) as the pruning algorithm. On HellaSwag, all sparsification methods incur only mild performance degradation, indicating that short-context and classification-style benchmarks are relatively robust to parameter removal. In contrast, GSM8K exhibits a markedly different behavior: performance degrades sharply as sparsity becomes more structured or aggressive. This phenomenon consistently appears across other language models, e.g., LLaMA-3 (Grattafiori et al., 2024) and Qwen-3 (Yang et al., 2025) (see Figure 10), reinforcing our claim that generation-oriented tasks impose stricter robustness requirements under network pruning.

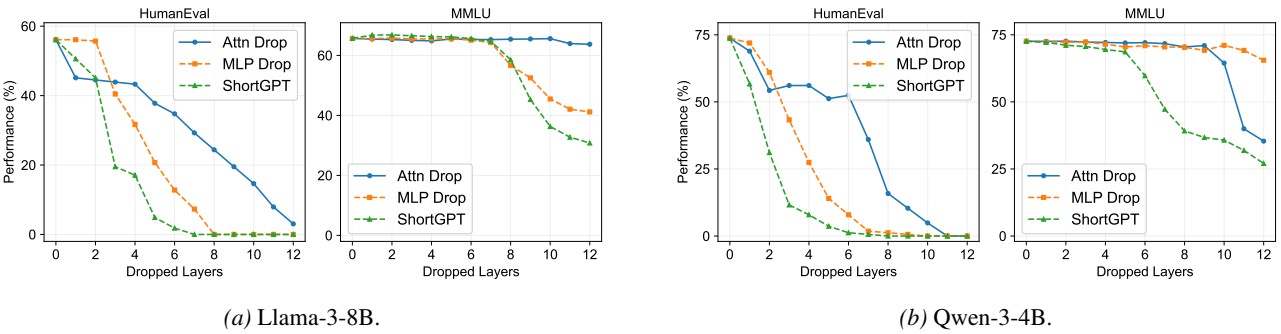

*Figure 10.* **Impact of inter-layer pruning on generative (HumanEval) and non-generative (MMLU) tasks**.

## H. Ablation Study on Temperature Factors

We conduct an ablation study on the temperature factor to examine the robustness of our analysis under different softmax scaling settings. Specifically, we vary the temperature while keeping all other configurations unchanged and compare the ground-truth measurements and theoretical estimates in terms of cosine similarity and KL divergence. As shown in

Figure 11, the estimated trends consistently align with the ground-truth measurements across different temperatures. These results indicate that our theoretical analysis is not sensitive to a particular temperature choice and generalizes well across commonly used temperature settings.

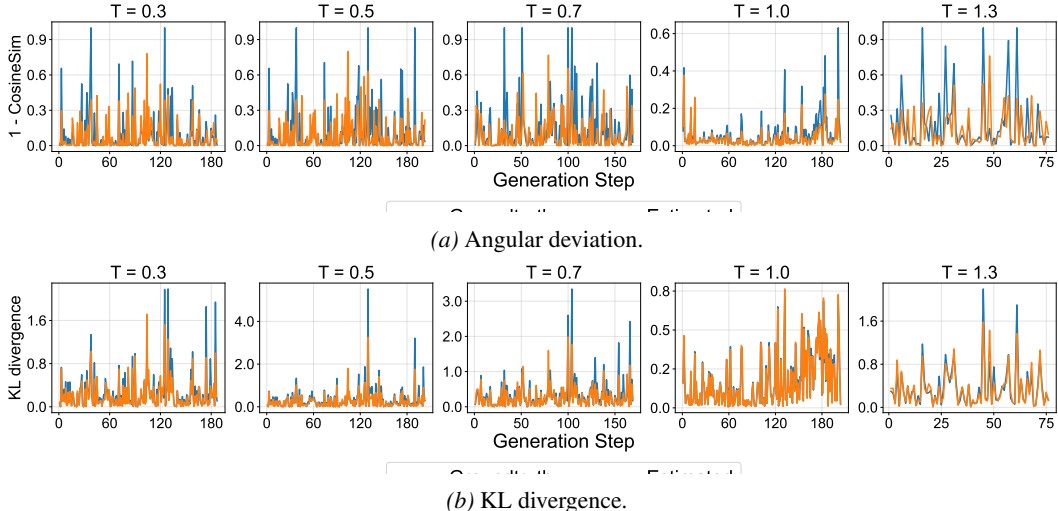

*(a)* Angular deviation.

*(b)* KL divergence.

*Figure 11.* **Effect of temperature on pruning-induced deviation estimation.** We compare the ground-truth measurements and theoretical estimates under different temperature settings in terms of (a) angular deviation and (b) KL divergence.

# I. Complementary Discussion of Quantization

While this work mainly focuses on network pruning, our empirical and theoretical analysis also applies to quantization. As shown in panels (a)–(c) of Figure 12, we compare the resulting deviations induced by quantization and pruning. Quantization exhibits consistently higher similarity, i.e., lower deviations, because it approximates parameters with low-precision values, whereas pruning removes parameters entirely. As a result, the magnitude and variance of $\Delta z$ are much lower, and the KL divergence of $p$ remains nearly stable in the early decoding steps. Although the KL divergence for quantization increases sharply at a certain point, this mainly occurs because the question has already been fully answered and redundant tokens are generated in the subsequent sequence.

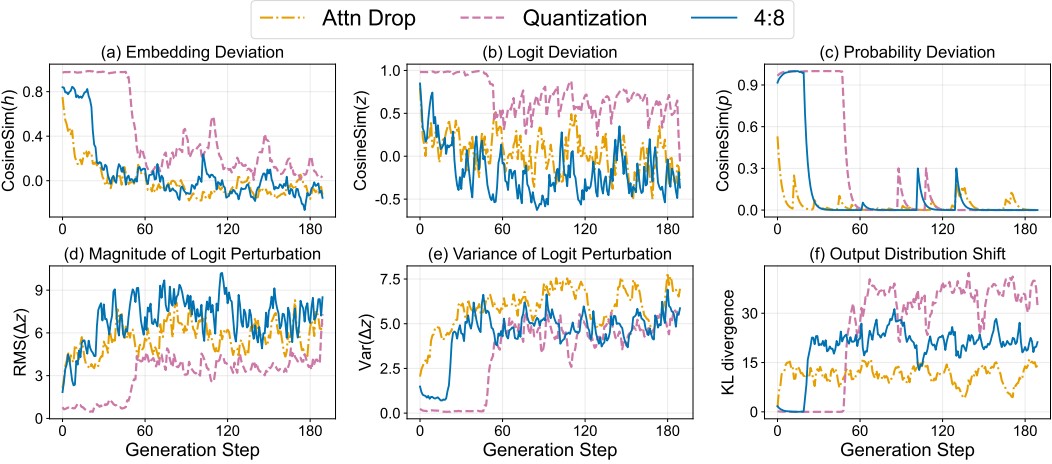

*Figure 12.* **Step-wise perturbation analysis under different compression methods.** We compare inter-layer dropping via Attn Drop with 8 layers removed, intra-layer sparsification via Wanda with 4:8 (50%) sparsity, and weight-only quantization using AWQ. Panels (a–c) report cosine similarity between the original and perturbed representations in the embedding, logit, and probability spaces, respectively. Panels (d–f) further characterize the magnitude of logit perturbations and their distributional effects, including the resulting KL divergence in the output probability distribution.

## J. Supplementary Visualizations

Beyond representation similarity induced by removing entire layers, we also examine the effect of intra-layer pruning. For the $i$-th layer, we prune that layer to a target sparsity level and measure the representation similarity between the outputs of the baseline model and the pruned model. Figure 13 shows that Wanda pruning follows a trend similar to that observed with layer dropping. While the magnitude differs across pruning strategies, e.g., MLP layers exhibit greater representation similarity after intra-layer pruning, the same representation-hierarchy interpretation still applies.

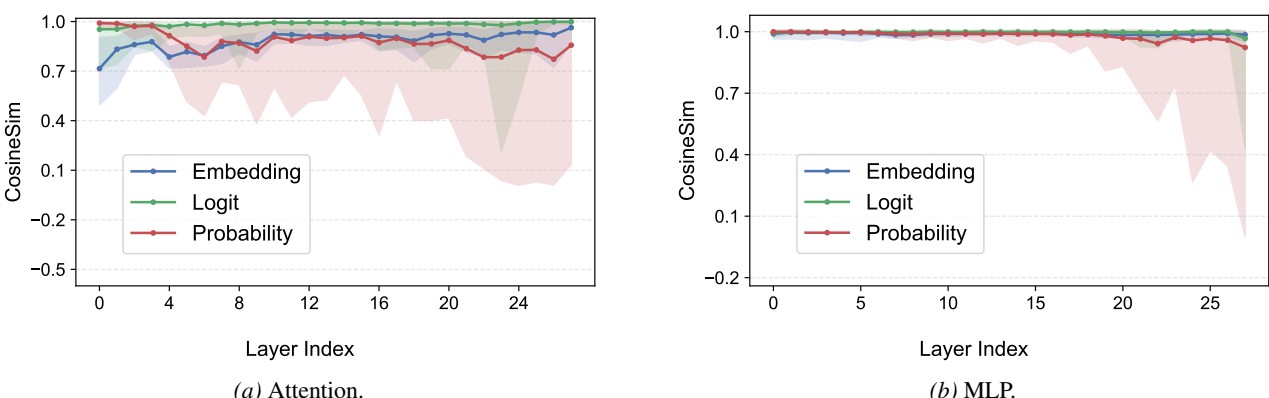

*(a)* Attention.                                                                 *(b)* MLP.

*Figure 13.* **Layer-wise representation similarity between the outputs of the baseline model and its temporarily pruned counterpart**. For the pruned model, we temporarily prune a single layer during the forward pass and compute the representation similarity at the same layer, while keeping all other layers identical to the baseline model.

We also present layer-wise comparisons between ground-truth measurements and theoretical estimates across different representation spaces to further validate the proposed approximation and trace how pruning-induced perturbations evolve during generation.

Figures 14 and 15 report the layer-wise evolution of distributional deviations in the probability space, measured by KL divergence and angular deviation, respectively. For each attention layer, we compare the ground-truth measurements with our theoretical estimates across decoding steps. The results show that the proposed estimator closely tracks the true deviation trends across layers and time steps. Notably, deeper layers consistently exhibit larger deviations, indicating stronger distributional shifts in later stages of the network.

Figures 16 and 17 further examine representation deviations in the embedding and logit spaces. In contrast to the probability space, both spaces show substantially smaller angular deviation, and the theoretical curves remain closely aligned with the ground-truth measurements. This observation supports our analysis that pruning-induced perturbations remain localized in these spaces and are less amplified before the softmax transformation. The only exceptions are the first and last layers, where the transformations are substantially larger and therefore violate the assumption of locality.

Finally, Figure 18 compares the relative magnitude ratios of representations in the embedding and logit spaces. The results indicate that the relative orthogonal energy is substantially reduced after the LM head projection, which is consistent with pruning-induced perturbations remaining limited in the logit space. Figure 19 shows the variance of $\Delta z$ under uniform and weighted sampling, while Figures 20 and 21 compare this variance against the corresponding relative magnitude ratios. The consistently large variance of $\Delta z$ explains the substantial deviation observed in the probability space.

Together, these visualizations corroborate the theoretical findings in the main text, illustrating how pruning-induced perturbations are progressively amplified across representation spaces, and highlighting the distinct roles played by linear and nonlinear transformations in this process.

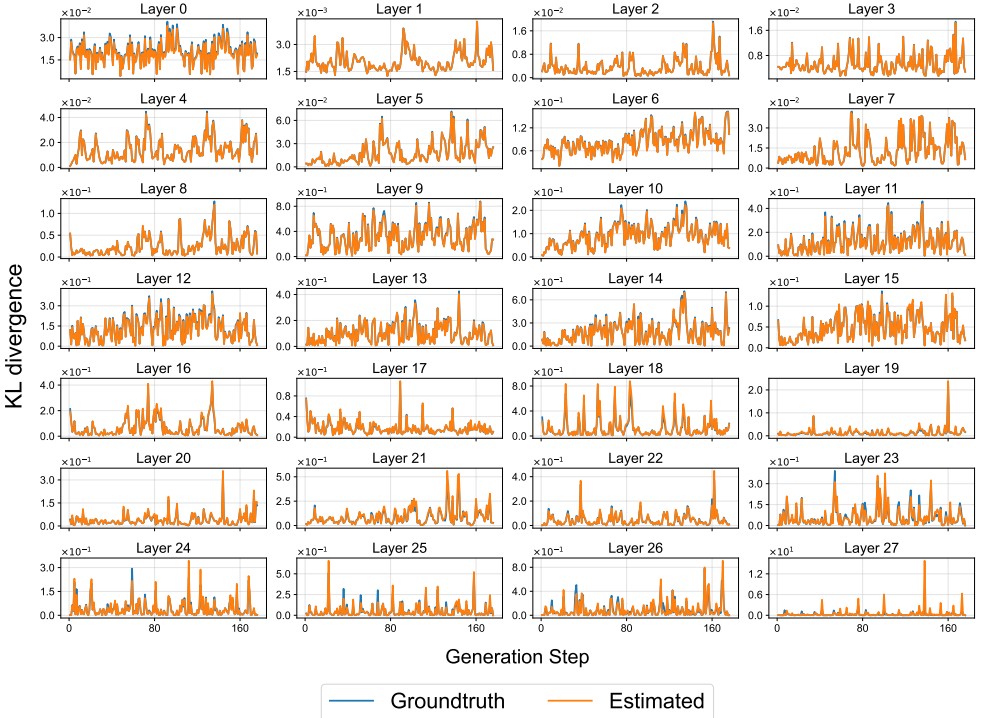

*Figure 14.* **Layer-wise evolution of KL divergence in attention layers across generation steps.** For each transformer layer, we compare the ground-truth KL divergence (blue) and the theoretical estimates (orange). The results show that the proposed estimator closely tracks the true KL behavior across layers, while also revealing that deeper layers generally exhibit larger distributional shifts.

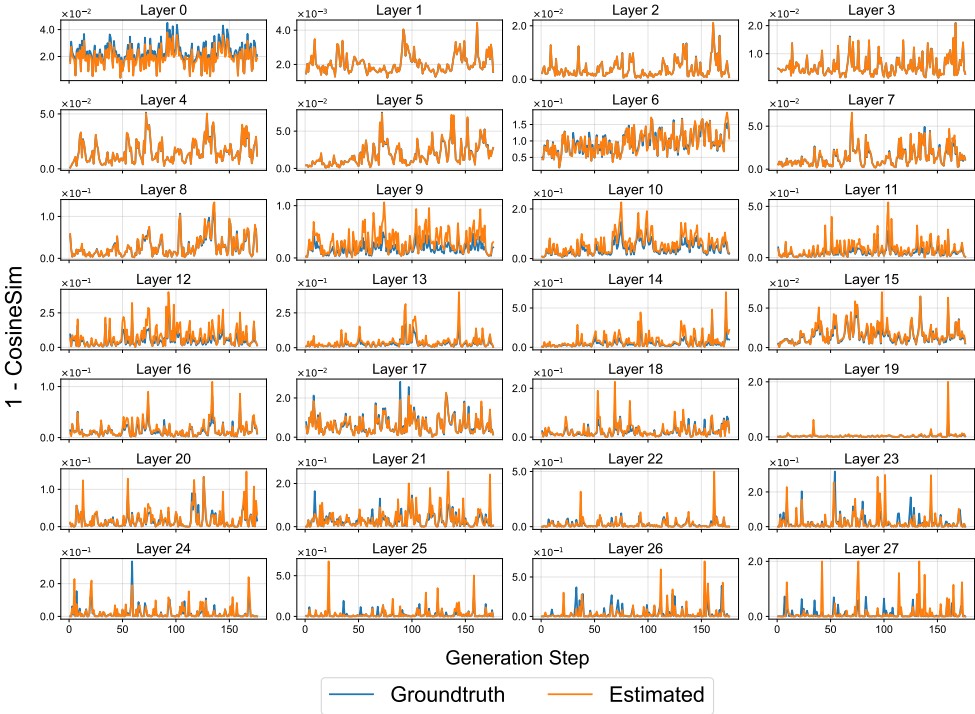

*Figure 15.* **Layer-wise evolution of angular deviation values** of the ***probability space*** between the inputs and outputs of attention layers. We report the ground-truth measurements (blue) and the theoretical estimates (orange) across decoding steps. Consistent with the KL analysis, shallow layers remain relatively stable while deeper layers exhibit larger semantic deviations, and our estimator successfully captures these trends.

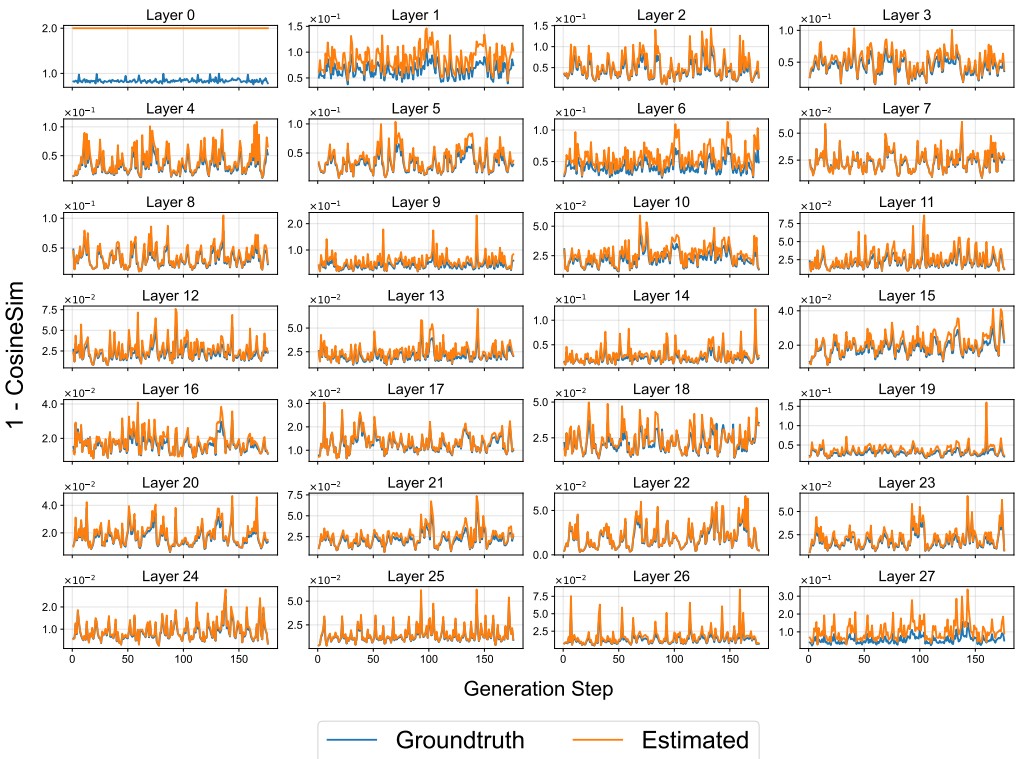

*Figure 16.* **Layer-wise evolution of angular deviation values** on the ***embedding space*** between the inputs and outputs of attention layers. We report the ground-truth measurements (blue) and the theoretical estimates (orange) across decoding steps.

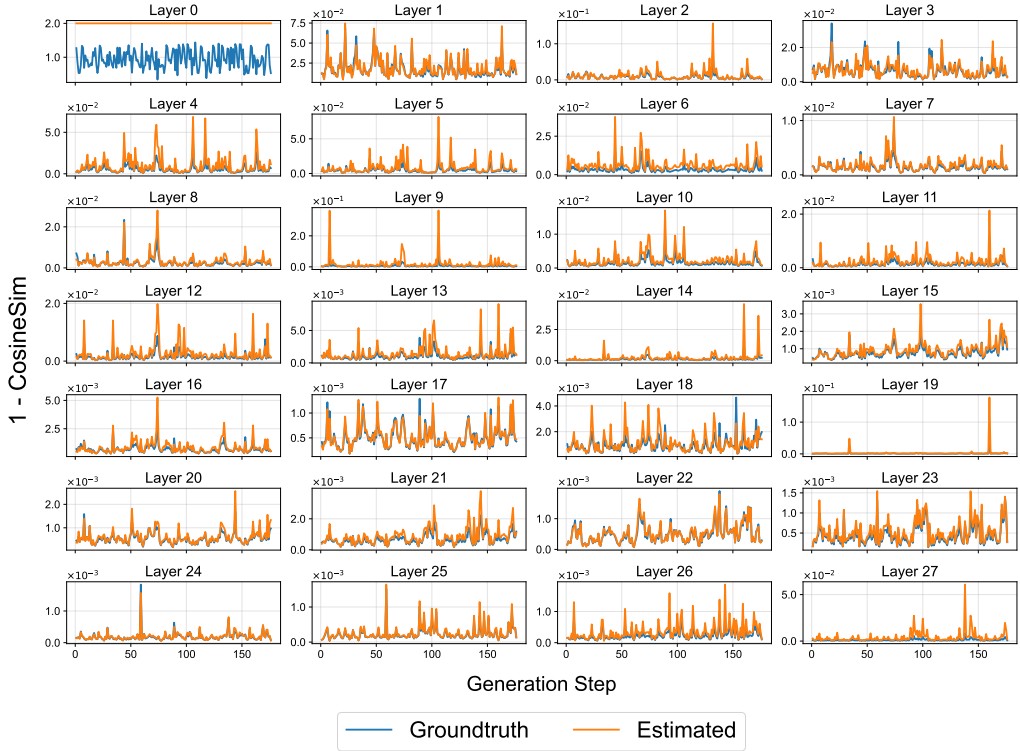

*Figure 17.* **Layer-wise evolution of angular deviation values** on the ***logit space*** between the inputs and outputs of attention layers. We report the ground-truth measurements (blue) and the theoretical estimates (orange) across decoding steps.

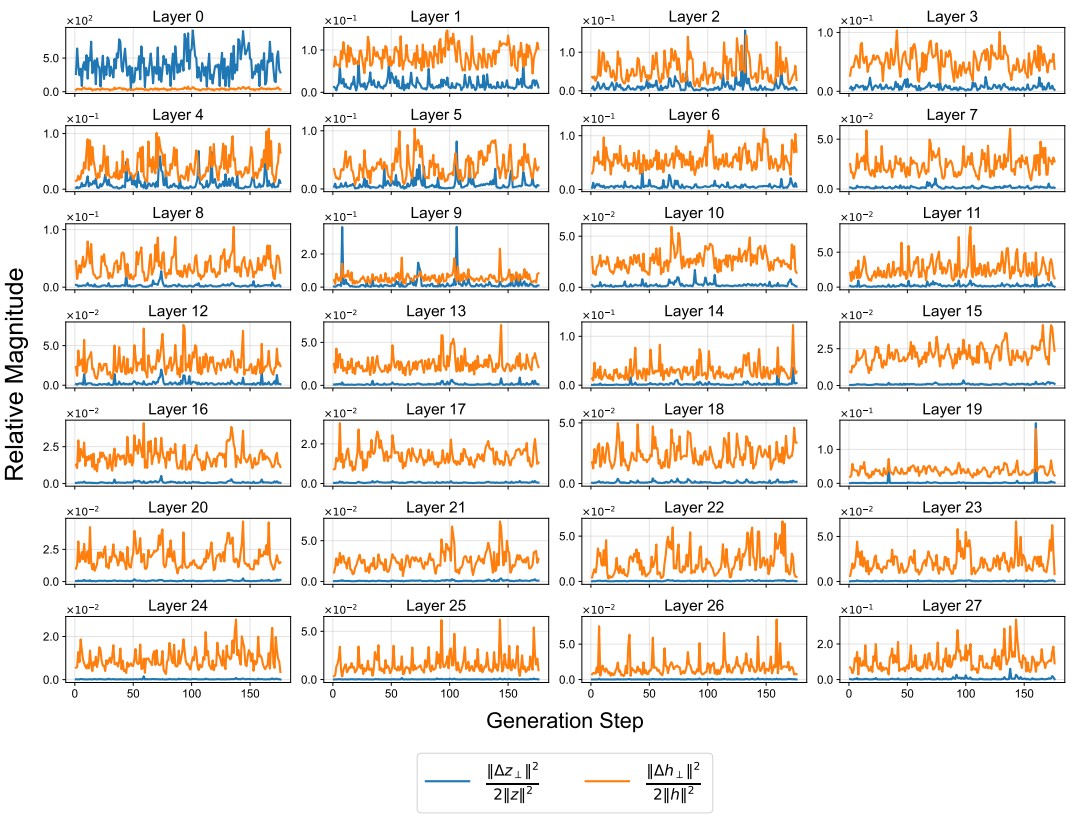

*Figure 18.* Relative magnitude ratios of representations in the ***logit space*** (blue) and the ***embedding space*** (orange).

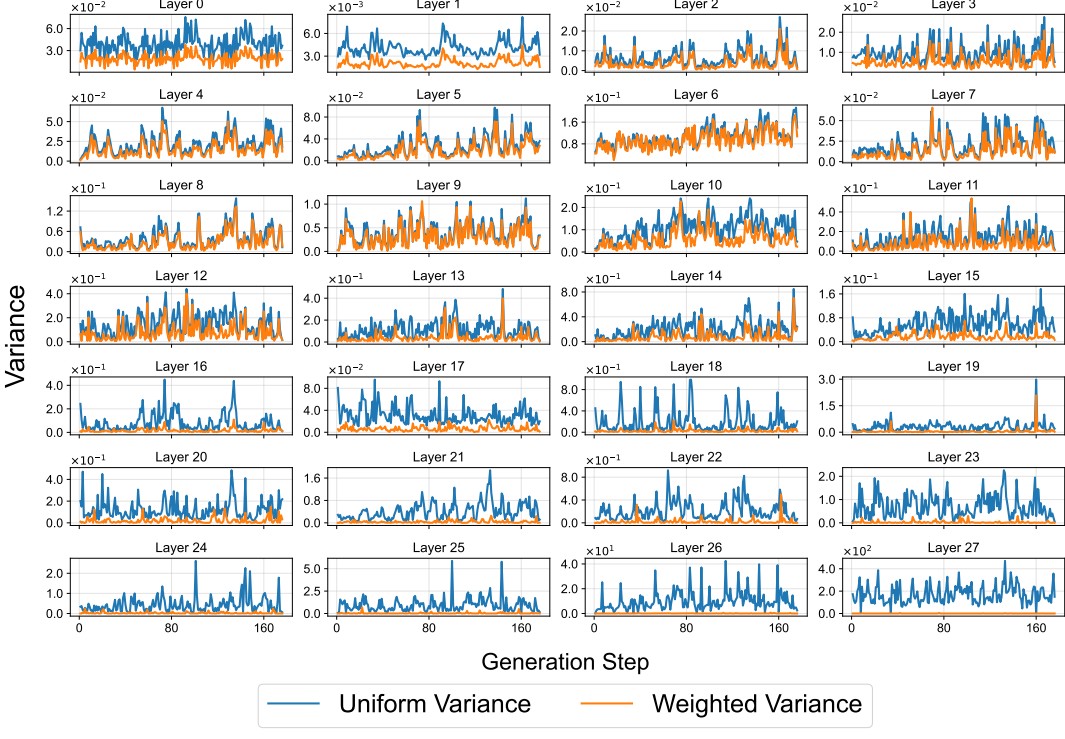

*Figure 19.* **Uniform and weighted variances of** $\Delta z$, where the uniform variance is computed under a ***uniform distribution*** (blue) and the ***weighted variance*** (orange) is computed under the vocabulary distribution $r_i = \frac{p_i^2}{\|p\|^2}$.

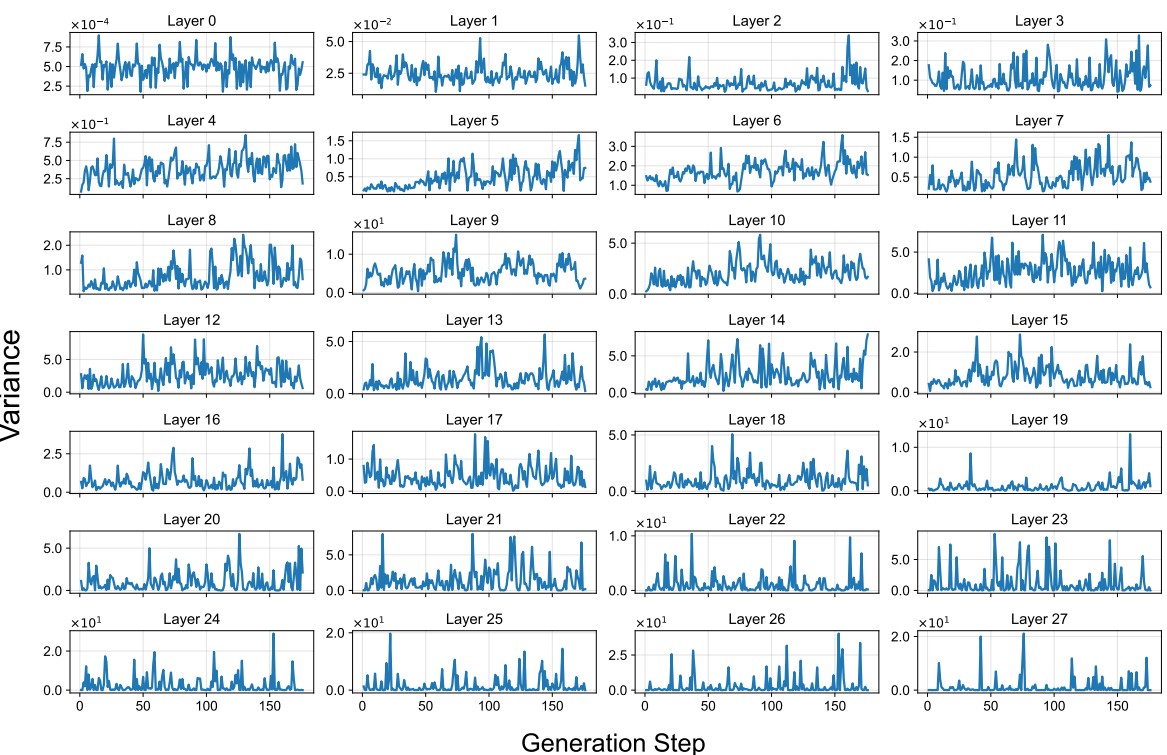

*Figure 20.* Relative values of the weighted variance of $\Delta z$, divided by the relative magnitude ratio of $h$.

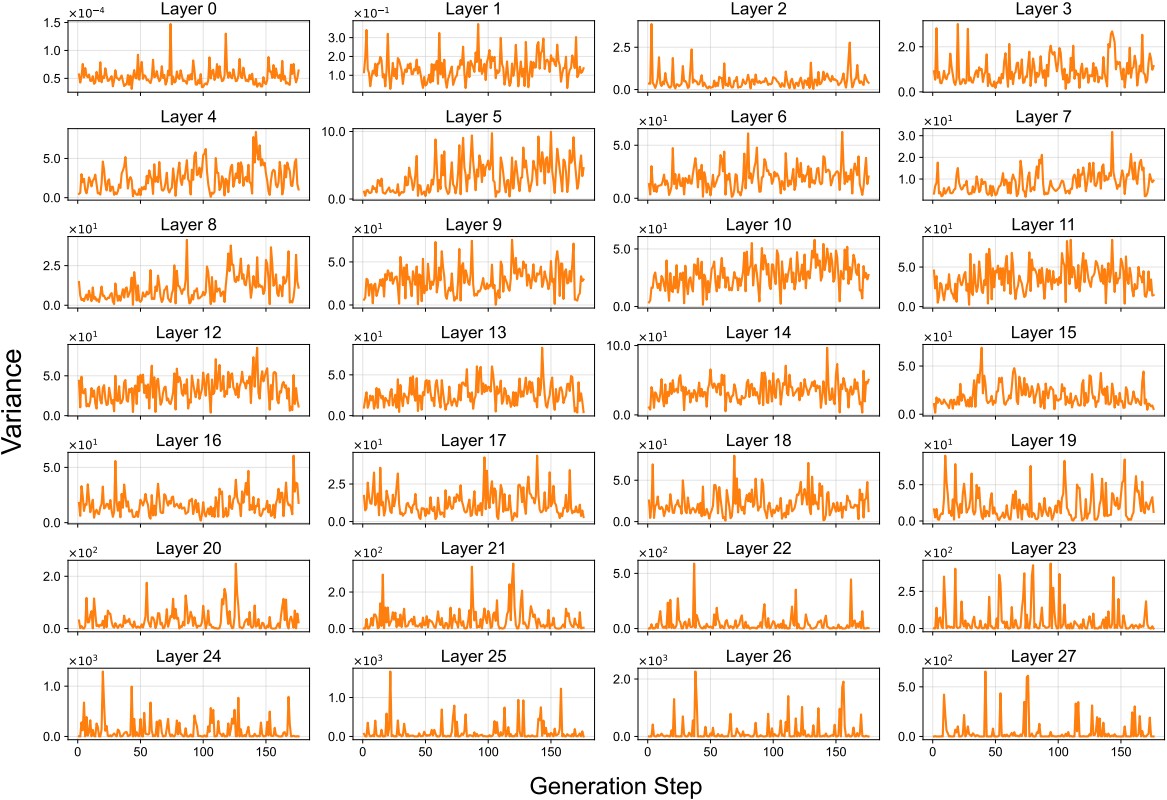

*Figure 21.* Relative values of the weighted variance of $\Delta z$, divided by the relative magnitude ratio of $z$.

