# OpenReview forum: "Demystifying When Pruning Works via Representation Hierarchies"
_ICML.cc/2026/Conference — ICML 2026 regular_

### Official Review · Reviewer_sVTe · 2026-03-09

**Soundness:** 3
**Presentation:** 3
**Significance:** 4
**Originality:** 3
**Overall Recommendation:** 4
**Confidence:** 4

**Summary:**

This work shows some analysis of the effectiveness of model pruning, especially for Large Language Models.
The author raises the observation that the pruned models perform diversely in generative and non-generative settings.
This work then analyzes pruning from the perspective of internal representation transformations in language models.
The experiments give evidence for the analysis.

**Compliance With Llm Reviewing Policy:**

Affirmed.

**Final Justification:**

My concerns are fully-addressed, I maintain the positive score.

**Key Questions For Authors:**

1. In Figure 4, how do you calculate the CosineSim of logit and probability for different layers, by directly inputting the activations into the LM head?
2. Did you find some interesting results that show difference between structured or unstructured pruning methods?
3. I am also curious about, whether other model compression methods, such as quantization, will show similar phenomenon.
4. If using the model's own output, combined with the prompt, as calibration data, will it be different?

**Limitations:**

See Weaknesses.

**Strengths And Weaknesses:**

--- Strengths

1. The observation of the divergence of performance between generative and non-generative tasks for pruned models is meaningful.
2. The analysis to study the phenomenon through perspective of internal representation is insightful and detailed.
3. The paper structure is clear and the claims are well-supported.

--- Weaknesses

1. Although the raised phenomenon is interesting, the paper shows these claims on a specific size of models, i.e. the 7B model. To better highlight the contributions, the authors are encouraged to test on smaller and larger size models.
2. For most of the pruning methods, the calibration data choice should be important to the final performace, but I didn't find the paper shows detailed discussion about it.
3. The scope of the model choice is limited on instruction model. Will the base model and reasoning model show similar phenomenon? For base model, it may help identify whether such mismatch is because of the instruct-tuning. For reasoning model, if such phenomenon still existed, on reasoning tasks it should degrade more than expected. By doing these, I think the analysis will be better.

---

> ### Author Rebuttal · Authors · 2026-03-31
>
> We thank the reviewer for the feedback and for recognizing the value of representation-level analysis.
>
> ### W1. Model scale coverage
>
> We conducted additional experiments on **Qwen3** models of different sizes under **Attention Drop**. The same qualitative pattern persists across all scales: **generative performance degrades much earlier and more severely than non-generative performance**.
>
> | Model | Metric | 0 layers | 4 layers | 8 layers | 12 layers |
> |---|---|---:|---:|---:|---:|
> | Qwen3-0.6B | GSM8K | 41.5 | 25.8 | 0.0 | 0.0 |
> | | MMLU | 47.4 | 46.8 | 33.5 | 23.7 |
> | Qwen3-1.7B | GSM8K | 68.2 | 36.3 | 0.0 | 0.0 |
> | | MMLU | 60.2 | 60.2 | 41.4 | 24.2 |
> | Qwen3-14B | GSM8K | 88.0 | 13.9 | 0.0 | 0.0 |
> | | MMLU | 78.9 | 78.9 | 69.1 | 57.9 |
>
> ### W2. Base and reasoning-style model coverage
>
> We also evaluated **base models** to test whether the effect is specific to instruction tuning. The same qualitative conclusion still holds: after pruning, **generative tasks degrade much more severely**, while non-generative tasks remain substantially more stable.
>
> | Model | GSM8K | HumanEval | RTE | ARC-Challenge | BoolQ | HellaSwag | MMLU | OpenBookQA | WinoGrande |
> |---|---:|---:|---:|---:|---:|---:|---:|---:|---:|
> | Qwen3-0.6B-Base | 49.7 | 28.0 | 71.5 | 44.5 | 69.9 | 53.6 | 52.4 | 35.0 | 61.0 |
> | Qwen3-0.6B-Base (4:8 Wanda) | 0.8 | 0.0 | 52.0 | 24.5 | 59.6 | 32.4 | 25.6 | 27.2 | 52.1 |
> | Qwen3-4B-Base | 74.7 | 56.1 | 76.5 | 64.6 | 82.8 | 75.7 | 73.0 | 40.8 | 72.1 |
> | Qwen3-4B-Base (4:8 Wanda) | 17.7 | 3.7 | 58.1 | 50.4 | 70.8 | 61.2 | 53.6 | 36.6 | 63.5 |
>
> Across **instruction-tuned, base, and reasoning-style models**, the same pattern consistently holds.
>
> ### W3. Calibration data
>
> Both settings use **128 C4 samples**. For **intra-layer pruning**, this follows the standard **Wanda** setup. For **inter-layer pruning**, we follow the **Attention Drop** setting and use **128 C4 samples**. In our experiments, the selected dropped layers remain relatively robust to the calibration choice [1].
>
> ### Q1. Details of Figure 4
>
> For each layer, we perform a **single-layer replacement**: we replace that layer in the dense model with its pruned counterpart while keeping all other layers unchanged, and then compare the resulting **embedding**, **logit**, and **probability** perturbations across decoding steps under the same dense-model-generated context. Figure 4 reports these layer-wise similarities under **layer dropping**, and the corresponding intra-layer analysis is provided in the appendix. We will make this pipeline clearer in the main text.
>
> ### Q2. Structured vs. unstructured pruning
>
> The qualitative finding holds for both settings (Figures 3 and 10): generative tasks are consistently more fragile. The key practical difference is that **unstructured pruning** introduces smaller perturbations and milder degradation, but provides **little to no realized efficiency gain** without dedicated sparse-kernel support. **Structured pruning** (layer drop, 2:4/4:8) delivers real speedups but causes more pronounced degradation. From a deployment standpoint, unstructured pruning's better performance is therefore somewhat moot—the practical risk lies in the structured settings where speedup is actually realizable.
>
> ### Q3. Connection to quantization
>
> Thank you for this question. We also applied the same representation-hierarchy analysis to **AWQ quantization** [2]. Quantization induces substantially smaller perturbations than pruning, and this lower-error regime persists for more decoding steps. This suggests that the same analysis framework may be useful more broadly for studying compression-induced perturbations beyond pruning.
>
> ### Q4. Self-generated calibration data
>
> We compared original C4 against **model-rewritten C4**. For **inter-layer pruning**, across **12 drop counts × 3 drop types × 2 model families**, only **1–2 cases** differ in the selected dropped layers. For **intra-layer pruning**, performance differences remain small (within 1–2 points on all benchmarks), as shown below for a representative model:
>
> | Model | GSM8K | HumanEval | RTE | ARC-C | BoolQ | HellaSwag | MMLU | OpenBookQA | WinoGrande |
> |---|---:|---:|---:|---:|---:|---:|---:|---:|---:|
> | Qwen3-4B | 85.0 | 72.0 | 75.8 | 62.4 | 85.1 | 70.0 | 70.2 | 40.0 | 66.7 |
> | + 4:8 Wanda (C4) | 21.5 | 4.2 | 74.7 | 46.1 | 78.9 | 54.3 | 48.6 | 34.4 | 60.1 |
> | + 4:8 Wanda (C4-rewrite) | 21.7 | 3.7 | 74.0 | 47.4 | 76.9 | 53.7 | 49.5 | 34.2 | 59.7 |
>
> ### Limitations
>
> This work focuses on **training-free pruning**. A key takeaway is that pruning can become unreliable for **generative tasks**, even under **moderate pruning**. Systematically studying **post-pruning training or adaptation** is a future direction not included in the current paper.
>
> **Reference**
>
> [1] Shwai He et al. *Uncovering the Redundancy in Transformers via a Unified Study of Layer Dropping*, TMLR.
>
> [2] Ji Lin et al. *AWQ: Activation-aware Weight Quantization for LLM Compression and Acceleration*, MLSys 2024.

---

> > ### Author Rebuttal · Reviewer_sVTe · 2026-04-03
> >
> > Thank you for your detailed clarification which resolves my concerns.
> > And your additional experiments show larger potential of the analysis.
> > I will not ask for more experiments, but only hope that the author will consider enrich the corresponding experiments in the revision. (Especially the calibartion part.)
> > I will keep my positive socre.

---

> > > ### Author Response · Authors · 2026-04-03
> > >
> > > **Update** [12:15 pm AoE Time April 7]: We sincerely thank the reviewer for the thoughtful follow-up and for confirming that the concerns are fully addressed. We are also very grateful for the maintained positive score. We will incorporate the helpful suggestions, especially on enriching the corresponding experiments and the calibration part, into the next version.
> > >
> > > ---
> > >
> > > Thank you very much for this highly encouraging follow-up. We sincerely appreciate your careful reading, thoughtful questions, and professional feedback throughout the discussion. We are especially grateful that you recognize the value of the representation-level analysis and the broader potential of this line of study.
> > >
> > > We also truly appreciate your positive assessment of the paper’s clarity and empirical support. Your suggestion on further enriching the corresponding experiments is very helpful, and we will incorporate it in the camera-ready revision.
> > >
> > > Thank you again for your time, support, and professionalism.

---

### Official Review · Reviewer_5EwS · 2026-03-10

**Soundness:** 3
**Presentation:** 3
**Significance:** 3
**Originality:** 3
**Overall Recommendation:** 5
**Confidence:** 4

**Summary:**

This paper studies why large language models fail at generation tasks when pruned, while still retaining good performance at non-generation tasks. Via experiments, they show that the deviations that emerge in the probability outputs via the non-linear softmax function are compounded by the iterative process of token generation, leading to collapse in the final output.

**Compliance With Llm Reviewing Policy:**

Affirmed.

**Final Justification:**

I recommend an accept.

The paper tackles a relevant problem related to post-training pruning without finetuning. As models become larger and larger, techniques to limit their size are required. By showing the difference in impact on non-generative and generative tasks, the door is open for pruning techniques that can limit this impact, leading to more efficient pruning. As such, I believe this paper can have an impact, and inspire both follow-up methods and analysis. The main point holding it back from a higher score is that -- while the observations are very clear -- the practical insights are slightly lacking, so I chose to not increase my assessment w.r.t. my original review.

**Key Questions For Authors:**

- What would happen if only a moderate amount of layers / parameters are removed? E.g. only 2 attention / MLP layers? Is the degradation still present?

- In the case of multiple-choice tasks that have categorical options, how is the output vocabulary limited to only these few options?

**Limitations:**

yes

**Strengths And Weaknesses:**

**Strengths**
- The paper presents the argument well, going in detail on the problem statement, with nice visualizations and clear language, leading to a very clear and comprehensible paper.

- The authors employ a large variety of models and tasks to illustrate the main problem associated with pruning-induced deviations.

**Weaknesses**

*Presentation*
- There is a small notational error in [Line 315], "delta_z orthogonal to z", this sentence handles the embedding space rather than the logit space, so should be "delta_h orthogonal to h".
- While it is clear for inter-layer pruning how much is removed, for intra-layer pruning this is not (e.g. figure 3).
- Several references are made to figures in the appendix (Figures 16, 17, 18 on page 6), and they are discussed in the main text. If these figures are necessary to understand the story of the paper, they should be included in the main text.

*Generalization.*

- More results on intra-layer pruning. Intra-layer pruning is is briefly mentioned and the impact of such pruning is discussed on the non-generative and generative tasks in table 3. However, all of the follow-up analysis with cosine similarity is only conducted on layer drop. Is there a specific reason for this? Could you show that the same observations hold for intra-layer pruning?

*Applicability.*

- The authors claim in [Lines 437-438, col 2] to offer 'practical insights and guidance for effective application of network pruning. While the authors shed a light on why this phenomenon happens, no practical advice is given on how to perform effective pruning, and mitigate this error accumulation

---

> ### Author Rebuttal · Authors · 2026-03-31
>
> We thank the reviewer for the positive assessment. We are encouraged that the reviewer finds the paper clear and comprehensible, and that the overall argument is well supported by the visualizations and broad empirical evaluation. Below we respond to the main points and clarify the corresponding revisions.
>
> ### W1. Notation issue
>
> Thank you for catching this. We agree that line 315 should refer to Δh and h, not Δz and z, because that sentence discusses the embedding-space perturbation.
>
> ### W2. Amount of pruning for intra-layer pruning
>
> We agree this should be much easier to identify. In the current experiments, the intra-layer settings correspond to **50% sparsity**, including **unstructured pruning**, **4:8 sparsity**, and **2:4 sparsity**. These settings are already described in the current draft, but we will state the exact sparsity level more explicitly in each relevant caption and paragraph.
>
> ### W3. Dependence on appendix figures
>
> This is a helpful suggestion. Our intention is that the main text already contains the core argument, while the appendix provides supporting plots and more detailed validation. That said, we agree the current narrative leans too much on appendix references in a few places. We will reduce that dependence and move the most important supporting figure(s) into the main story where possible, rather than relying on appendix references for the main narrative.
>
> ### W4. Main analysis centers on layer drop
>
> We use **layer drop** as the primary setting for the main mechanism analysis because it is more compelling from a practical efficiency perspective: removing entire layers can translate more directly into realized acceleration, whereas intra-layer sparsity often depends heavily on hardware and kernel support.
>
> Importantly, we do **not** intend to claim that the mechanism is unique to layer drop. The appendix already shows the same qualitative pattern for intra-layer pruning: relatively small perturbations in embedding/logit spaces but much larger deviations in probability space. We emphasize layer drop in the main text because it is the clearest practically relevant setting for presenting the mechanism.
>
> ### W5. Evidence for intra-layer pruning
>
> We agree this should be highlighted more directly. The current draft already includes supporting intra-layer evidence in **Figure 9** and **Figure 13**. In particular, **Figure 9** shows the step-wise perturbation behavior under different compression strategies, while **Figure 13** provides representation analysis for intra-layer pruning. These results show the same overall pattern as layer drop: perturbations remain relatively limited in the **embedding** and **logit** spaces, while the deviation becomes much more pronounced in the **probability** space. We will bring this out more clearly in the revision.
>
> ### W6. Practical insights and guidance
>
> Thank you for flagging this wording. We will make the claim more concrete. The practical guidance is not that the paper introduces a slightly better training-free pruning rule, but that it identifies **where the current pruning paradigm appears safe and where it is risky**. Our main practical takeaway is:
>
> **training-free pruning is much safer for single-step non-generative tasks than for open-ended generation, and deployment in the latter regime likely requires post-pruning adaptation.**
>
> ### Q1. Moderate pruning
>
> We agree that moderate pruning is an important case. The degradation is less severe under milder pruning, but the efficiency gains are also smaller. More importantly, even seemingly moderate pruning can already cause large failures in generation. As shown in **Figure 1**, dropping just **two** Transformer blocks or MLP layers reduces **HumanEval** by more than **20 points**, while non-generative tasks remain comparatively stable. Pruning attention layers is less destructive, but dropping **four** attention layers already leads to severe generation collapse.
>
> ### Q2. Multiple-choice output restriction
>
> We would like to clarify that the model still produces logits over the full vocabulary, but evaluation is performed by extracting the probabilities corresponding to the candidate-option tokens and taking the **argmax within that candidate set**. Thus, the decision depends only on a small candidate-token subspace such as **A/B/C/D**, rather than on the full output distribution. We will make this protocol easier to identify in the revision.
>
> ### Limitations
>
> This work focuses on **training-free pruning**. A key takeaway is that pruning can become unreliable for **generative tasks**, even under **moderate pruning**. A natural next step is to study **post-pruning training or adaptation** systematically, which is not included in the current paper. We will clarify this scope more explicitly in the revision.

---

> > ### Author Rebuttal · Reviewer_5EwS · 2026-04-03
> >
> > Dear author(s),
> >
> > Thank you for your extensive rebuttal. While you have answered most of my concerns and questions, I am still stuck on the *practical insights and guidance*. The current manuscript shows that generation tasks are more at risk for pruning than non-generative tasks, so you extract the practical insight that "*training-free pruning is much safer for single-step non-generative tasks than for open-ended generation, and deployment in the latter regime likely requires post-pruning adaptation.*" If find that quite a weak insight, and much rather would have some extended discussion on pruning rates, or the amount of generation steps. As such, unless these are addressed more in detail, I will stay at my original assessment.

---

> > > ### Author Response · Authors · 2026-04-03
> > >
> > > **Update** [8:45 am AoE Time April 7]: **We are truly grateful for your thoughtful final justification and recommendation for acceptance.** We are especially encouraged by your recognition that this work can inspire follow-up methods and analysis, and holds meaningful impact for the community. We will incorporate your feedback on practical guidance into the camera-ready version.
> > >
> > > ---
> > >
> > > Thank you for the follow-up and for pointing out that the practical guidance should be made more explicit. This is an important point, and below we provide concrete guidance, including the dimensions you mentioned: **pruning rates** and **generation steps**.
> > >
> > > First, **non-generative tasks remain much more robust under pruning**, and in this regime pruning can still be practically useful. A practical and commonly used procedure is to **first set a target performance threshold on validation data** (e.g., preserving about **95%** of the original performance), and then **gradually increase the pruning ratio until reaching the largest sparsity level that still satisfies this threshold**. Prior layer-dropping results show that one can remove up to *12/32 attention layers in Llama-3-8B* or *40/80 attention layers in Llama-3-70B* while still retaining about 95% of the original non-generative performance [1].
> > >
> > > Second, for generative tasks, **direct deployment of training-free pruned models is not advisable**. In our results, even mild or moderate pruning can already damage generation substantially: for example, dropping only two Transformer blocks or MLP layers reduces HumanEval by more than 20 points, and dropping four attention layers also leads to generation collapse. At the same time, the corresponding efficiency gain is often less than 10%, so the efficiency-accuracy tradeoff is already unattractive for generative deployment.
> > >
> > > Third, regarding generation steps, **the generation degradation cannot be mitigated simply by restricting the number of generation steps**. Even the simplest generative tasks with very short outputs already fail severely under pruning. We verified this in a controlled arithmetic setting on 48 simple prompts such as `7 + 5 =`, with generation capped at 8 tokens. The dense model reaches 97.9% accuracy, while Drop-8MLP falls to 16.7% and Drop-8Attn to 10.4%. This shows that the gap is not explained by long generation: pruning already induces large errors at the very first decoding step, and longer decoding mainly compounds a failure that has already emerged. This is also consistent with Table 2, where pruned models already exhibit incoherent generation, repetition, and collapse within only a few decoding steps. Therefore, **deploying training-free pruned models on generation tasks is fundamentally unreliable, even when the generation length is very short**.
> > >
> > > Finally, to preserve generation performance after pruning, **post-pruning training or adaptation is necessary to recover the degraded generation capability**. In such cases, we recommend an **iterative prune-and-adapt** strategy: in each iteration, **increase the pruning ratio by a small step** (e.g., ~5%) and **train to recover generation capability**; keep track of the last satisfying model, and exit when performance falls below the acceptable threshold, returning to the last satisfying pruned model.
> > >
> > > We sincerely thank the reviewer for the constructive suggestion. We have worked to develop the above guidance and hope it enhance this work. We warmly welcome any further feedback, and we commit to incorporating these recommendations more prominently in the next version.
> > >
> > > Reference
> > >
> > > [1] He et al. *Uncovering the Redundancy in Transformers via a Unified Study of Layer Dropping*, TMLR.

---

### Official Review · Reviewer_zf7e · 2026-03-10

**Soundness:** 2
**Presentation:** 2
**Significance:** 3
**Originality:** 3
**Overall Recommendation:** 4
**Confidence:** 3

**Summary:**

This work investigates pruning in transformer language models, specifically why pruning hurts text generation more than non-generative tasks. It traces how seemingly-similar representations from before and after pruned layers led to larger deviations in probability space, which aggregates errors across multiple generation steps.

**Compliance With Llm Reviewing Policy:**

Affirmed.

**Final Justification:**

The rebuttal has addressed my concerns, thus I increased my score to positive.

**Key Questions For Authors:**

- How do you see this work connecting to Logit Lens?
- It seems that, across generation steps, the deviation sometimes fluctuates a lot -- do you have insight to what kind of tokens are hampered more by pruning vs. others during generation?
- If the categorical choice tokens are not among the top tokens, it also seems that non-generative tasks are more protected from pruning because they deviate more from the next-token prediction objective.

**Limitations:**

- The paper is quite light on discussing the limitations of the work.

**Strengths And Weaknesses:**

Strengths:
- I think the work is timely
- The work lays out an intuitive hypothesis of why pruning hurts generation and provides useful evidence for it.
- The authors investigate the degradation from pruning from multiple angles rather than relying on a single metric.

Weakness:
- The paper mentions to study both intra- and inter-layer pruning, but iiuc most analysis focus more on pruning entire layers. It should be made more clear how effects from both pruning strategies are disentangled.
- I think the fact that these models perform much worse on the generative tasks to begin with complicates these investigations quite a bit -- e.g., the solutions were already brittle or there are too few computational paths to solve the task, leading them to be less protected from pruning. I think some efforts to balance the "task difficulty" could greatly enhance the paper.
- I find that the overall clarity of the work could be significantly improved. Specifically:
    - Some introduction of Wanda/LayerDrop is needed for a broader audience.
    - It's often unclear what pruning setting is studied for a particular analysis. E.g., in 6.1/figure 5, are the effects estimated from pruning a particular layer or averaged over layers?
    - Line 316 seems it should be referencing h instead of z.
    - "Feature space" is undefined.

---

> ### Author Rebuttal · Authors · 2026-03-31
>
> We thank the reviewer for the thoughtful feedback and constructive questions.
>
> ### W1.  Inter-layer vs. intra-layer emphasis
>
> We would like to clarify this distinction more explicitly. Our main-text emphasis on **inter-layer pruning** is intentional because it is the more practically relevant setting: dropping whole layers yields realized speedups much closer to the theoretical reduction, while many intra-layer sparsity methods still rely on specialized kernels for wall-clock gains.
>
> At the same time, the phenomenon is **not specific** to layer dropping. The appendix already includes supporting evidence for **intra-layer pruning**, including **SparseGPT** task results, **Figure 13** for representation analysis, and step-wise perturbation comparisons across compression strategies. We will make this cross-setting consistency more explicit in the revision.
>
> ### W2. Clarifying the pruning setup in each analysis
>
> We would like to clarify that these settings are already specified in the current draft, though they can be made easier to identify. In particular, the analyses in **Section 6.1 / Figure 5** are conducted under **layer dropping**, since this part follows the preceding observation under the same setting and keeps the mechanism analysis consistent. For **intra-layer pruning**, we keep the sparsity level fixed at **50%**, varying only the sparsity pattern (e.g., **unstructured**, **4:8**, **2:4**). We will revise the captions and nearby text to make this more explicit.
>
> ### W3. Task difficulty / brittleness
>
> We would like to clarify that the central issue is the much larger deviation that pruning induces in the **probability space**, rather than task difficulty itself. Under the **same input sequence**, changing only **one layer** can keep the embedding and logit spaces relatively stable while already producing a much larger deviation in the **probability** space. This means the discrepancy can emerge **before** long-horizon histories diverge. **Figure 8** directly illustrates this: pruning substantially distorts the **top-token distribution** even under fixed context. **Table 2** further shows generation collapse in practice, with incoherent outputs appearing within only a few decoding steps.
>
> The arithmetic control provides additional support. On 48 simple prompts such as *"`7 + 5 =`"*, capped at 16 tokens, the uncompressed model achieves **100% accuracy**, while **Drop-8MLP** and **Drop-8Attn** fall to **16.7%** and **10.4%**. This suggests that the observed gap is driven primarily by pruning-induced probability-space distortion, rather than by task difficulty alone.
>
> ### W4. Clarity, background, and notation
>
> We will improve the exposition by:
>
> - add brief background on **Wanda** and **LayerDrop**;
> - state the pruning setting more explicitly in each figure/caption;
> - fix the notation issue around line 316;
> - replace the vague term **feature space** with the more precise **representation space**.
>
> ### Q1. Connection to Logit Lens
>
> Both Logit Lens and our work analyze intermediate representations, but they answer different questions. Logit Lens probes what intermediate layers already predict, whereas our work studies how **pruning-induced perturbations** evolve across the **embedding, logit, and probability** spaces and why this affects task regimes differently. Our focus is not layer-wise interpretability itself, but the mechanism behind pruning robustness and failure.
>
> ### Q2. Fluctuation across decoding steps / token sensitivity
>
> We agree that token type is one plausible source of the observed step-wise fluctuation. Our preliminary inspection suggests that when the dense and pruned models assign high probability to the same shared formatting or special tokens, such as punctuation, spacing-related delimiters, or other low-information tokens, the observed divergence at that step is often relatively small. By contrast, larger deviations tend to appear on semantically informative non-special tokens, where pruning is more likely to alter the relative ranking among competing continuations.
>
> Once divergence enters a sufficiently large regime, local fluctuations should not be interpreted as recovery, but as variation along two already-separated trajectories.
>
> ### Q3. Categorical choice tokens in non-generative tasks
>
> We appreciate this observation—it aligns with our **task-relevant subspace** interpretation. In multiple-choice settings, the decision depends only on a small subset of candidate tokens that often do not coincide with the top-probability tokens. Pruning can distort the full distribution while leaving this **candidate-token subspace** comparatively stable. We will make this explicit in the revision.
>
> ### Limitations
>
> This work focuses on **training-free pruning**. A key takeaway is that pruning can become unreliable for **generative tasks**, even under **moderate pruning**. Systematically studying **post-pruning training or adaptation** is a future direction not included in the current paper.

---

> > ### Author Rebuttal · Reviewer_zf7e · 2026-04-03
> >
> > I thank the authors for the additional clarifications and for engaging with these questions/concerns. I think some of these additional clarifications mentioned could enhance the clarity and presentation of the paper. However, I remain concerned on controlling for task difficulty. To clarify, the core issue isn't task difficulty per se, but the possibility that even the intact models have not learned equally robust solutions to generative vs. non-generative tasks. Thus the perturbation in probability space from pruning could be a downstream result from weaker internal circuits, rather than the direct cause of performance lost in less well-learned tasks. I thus intend to maintain my original assessment.

---

> > > ### Author Response · Authors · 2026-04-03
> > >
> > > **Update** [11:25 am AoE Time]: **We sincerely thank you for raising your rating and for your recognition of our work.** We will incorporate your feedback into the camera-ready version. We also greatly appreciate your professionalism and constructive suggestions, which have helped improve our work and will benefit the research community.
> > >
> > > ---
> > >
> > > Thank you for the follow-up and for clarifying your concern more precisely.
> > >
> > > **We believe the point you raise is closely aligned with, rather than contrary to, the paper's main findings and analysis.** More specifically, regarding the possibility that *"the intact models have not learned equally robust solutions to generative vs. non-generative tasks"*, we believe this is exactly consistent with our findings. In our analysis, the embedding and logit spaces indicate substantial robustness, whereas the deviation in the probability space becomes much larger. Correspondingly, the generative tasks that depend directly on the full-vocabulary probability distribution appear much more fragile, whereas non-generative tasks remain comparatively more robust. In this sense, your point is consistent with our main observation.
> > >
> > > Likewise, regarding the concern that *"the perturbation in probability space from pruning could be a downstream result from weaker internal circuits"*, this is also closely related to what we measure in the paper. We explicitly quantify pruning-induced perturbations in the probability space, including via cosine similarity and KL divergence, and analyze why the perturbation remains relatively small in the embedding/logit spaces but becomes much larger in the probability space. This is supported not only empirically but also theoretically, for example by Theorem 2 on the sensitivity of the probability space to logit perturbations. **Therefore, we view your point as complementary to, rather than in conflict with, the paper's main explanation.**
> > >
> > > **Our earlier discussion of task difficulty was intended precisely to reduce the possibility that the observed phenomenon is explained simply by task difficulty.** Even in very simple tasks, the pruned model can still collapse, which in turn supports our main point that the different robustness of the underlying representation spaces contributes more directly to the different effects of pruning across tasks.
> > >
> > > More broadly, we believe the paper systematically raises and explains this important pruning issue: why pruning can preserve non-generative performance reasonably well while still causing severe failures in generation, and how this difference can be understood through representation hierarchies and probability-space sensitivity.
> > >
> > > If you still have any remaining questions, we would be very happy to address them before the discussion deadline in the coming days.
> > >
> > > Update: We sincerely appreciate your careful consideration of our rebuttal and follow-up response, and thank you for your recognition of the value of our work in your updated rating.

---

### Official Review · Reviewer_HAQv · 2026-03-12

**Soundness:** 3
**Presentation:** 2
**Significance:** 2
**Originality:** 3
**Overall Recommendation:** 3
**Confidence:** 3

**Summary:**

This paper studies LLM pruning and empirically finds that the performance gap on generation tasks is significantly larger than on non-generation tasks. The authors categorize the effects of pruning into three different parts: embedding, logits, and probability, and find that pruning at the probability level degrades generation performance. The errors will accumulate in generation tasks and lead to a significant performance drop. Several theorems are introduced to explain these observations.

**Compliance With Llm Reviewing Policy:**

Affirmed.

**Final Justification:**

The rebuttal addressed several of my concerns, but my concerns regarding ``significance'' remain.  A few other technical questions are also not fully addressed.

The major concern is that this paper's contribution is mainly diagnosing, analyzing, and showing observations, without enough technical contribution beyond that. I also read other reviewers' feedback, and it seems that the reviewer 5EwS has a similar concern but gave positive feedback. If other reviewers think the contribution is sufficient for acceptance, I am happy to follow the overall assessment. Therefore, I'll update my score to a weak reject and lower my confidence level.

**Key Questions For Authors:**

1. See soundness 1. If the generative/non-generative tasks use the same model/architecture, I believe the major difference between them is the length of generated tokens. Naturally, generating more tokens will accumulate errors. Can the authors provide some analysis/ablations showing the relationship between the performance gap wrt the length of generation?

2. In Table 1, which pruning method do the authors use for this model?


3. Line 260-264, in the second column, "Notably, the logit space exhibits even higher ... from embedding to logits". Could the authors elaborate more?

4. In Figure 3, for the GSMK task, the performance drop of Qwen is not as large as that of Llama and Mistal. Actually, the performance drop of Qwen on the non-generative task is close to that of the generative task. Could the authors explain why this happened?

**Limitations:**

yes

**Strengths And Weaknesses:**

## Soundness

The paper argues that model pruning preserves non-generative performance but often degrades generation. However, since the same model and architecture are used, the major difference between non-generative and generative tasks may be the length of the generated tokens. Naturally, generating more tokens can accumulate errors, but the paper does not provide analysis or ablations to isolate this effect.

## Presentation
Overall, the presentation is easy to follow; the figures and tables are illustrative enough to help understand the concepts and claims. However, the paper does not follow a typical conference submission format and reads more like a technical report. Besides, there are several typos, including not changing the running title "Submission and Formatting Instructions for ICML 2026."

## Significance
The paper presents interesting empirical findings, but the community would benefit more if the authors could provide solutions to the issues that pruning degrades generation performance. For example, the authors could propose new pruning methods or algorithms that can mitigate this issue or explore ways to correct the resulting errors.

## Originality
The author provides new insights into how pruning affects different tasks. Specifically, the representation hierarchies offer a new and interesting perspective. The paper mostly focuses on empirical studies, and these empirical discussions are valuable.

I think this submission is overall a good starting point, but the current study seems incomplete.

---

> ### Author Rebuttal · Authors · 2026-03-31
>
> We thank the reviewer for the careful feedback.
>
> ### W1. Generation length vs. single-step mechanism
>
> We would like to clarify that the paper already isolates this effect more explicitly than the review suggests.
>
> Under **fixed context**, replacing only one dense layer with its pruned counterpart already produces large deviations in **probability space** while leaving embedding and logit spaces relatively stable (**Figure 4**, **Figure 6**). Under **free autoregressive generation**, this deviation is further amplified across time (**Figure 7**, **Figure 9**, **Appendix D**).
>
> To directly address the decoding-length concern, we ran a controlled experiment on 48 simple arithmetic prompts (e.g., *"`7 + 5 =`"*), capping generation at **8 / 16 / 32** tokens:
>
> | Method | Acc@8 | Acc@16 | Acc@32 |
> | --- | ---: | ---: | ---: |
> | Baseline (uncompressed) | 97.9 | 100.0 | 100.0 |
> | Drop-4Attn | 60.4 | 60.4 | 60.4 |
> | Drop-4MLP | 64.5 | 64.5 | 64.5 |
> | Drop-8Attn | 10.4 | 10.4 | 10.4 |
> | Drop-8MLP | 16.7 | 16.7 | 16.7 |
>
> The baseline's 97.9% at @8 reflects one verbose response exceeding the token limit; it recovers to 100% at @16. Across all **48** questions, these results confirm that the gap is **not explained by long generation**: pruning induces large errors **at the very first decoding step**, before any cross-step history diverges—autoregressive length then **compounds**, rather than causes, this initial failure. The failure is therefore fundamentally a **single-step representation failure**, not an error-accumulation artifact.
>
> ### W2. Mitigation, scope, and completeness
>
> We agree mitigation matters. We already discuss **post-pruning adaptation** as the natural recovery path, and our results suggest the gap is hard to resolve within the **training-free** paradigm—even mild pruning already causes severe generation failure.
>
> The paper's goal is therefore primarily **diagnostic**: identifying **where pruning remains reliable, where it breaks down, and why**. This is itself a practically useful contribution: **training-free pruning is much safer for non-generative settings than for open-ended generation**, and the latter likely requires post-pruning adaptation [1].
>
> The paper is complete with respect to this objective: it establishes the empirical discrepancy, identifies the representation-level mechanism, provides theoretical support for the **logit-to-probability amplification**, and separates single-step perturbation from autoregressive accumulation. We do point to a mitigation direction: **post-pruning adaptation**. However, given the severity and early onset of the failure, we argue that training-free remedies are hard to achieve—the right fix requires stepping **outside the training-free paradigm**. Identifying when and why this paradigm breaks down is a necessary precondition for designing effective mitigations.
>
> ### W3. Formatting and typos
>
> We will fix the template artifact, running-title issue, and remaining typos in the revision.
>
> ### Q1. Table 1 setup
>
> Table 1 already uses **LayerDrop-style inter-layer pruning**; **Drop-8A / Drop-8M** denote dropping **8 attention** or **8 MLP** layers. This is already specified in the current draft: **Lines 193-207** introduce inter-layer pruning and adopt **LayerDrop**, while **Lines 214-219** and the **Table 1** caption specify dropping **eight attention or MLP layers**. We will restate this more explicitly in the caption.
>
> ### Q2. High logit similarity
>
> The **embedding-to-logit** transformation is relatively stable under pruning—logit similarity can even be slightly higher than hidden-state similarity—because the linear projection reduces the **relative orthogonal component** of the perturbation, which mainly governs cosine deviation as shown in **Figure 5** and **Figure 18**. This holds for both pretrained and randomly initialized LM heads, confirming the discrepancy does **not** arise at this step. By contrast, the **logit-to-probability** mapping is nonlinear and amplifies small logit changes into much larger probability shifts. We will revise **Lines 260-264** to make this transition more explicit.
>
> ### Q3. Qwen in Figure 3
>
> The Qwen results still follow the same qualitative pattern. For **Qwen2.5-7B**, under **4:8** and **2:4** pruning, **HellaSwag** drops from **81.4** to **69.3 / 61.8**, while **GSM8K** drops more sharply from **75.6** to **54.0 / 36.2**. Thus, generation remains more fragile under pruning in Qwen as well. While explaining cross-model robustness differences requires a broader comparative study, the same **task-level** discrepancy still holds here.
>
> ### Limitations
>
> This work focuses on **training-free pruning**. Systematically studying **post-pruning training or adaptation** is an important direction for future work not covered here. We will revise the limitations section to state this scope more explicitly.
>
> **Reference**
>
> [1] Shwai He et al. *Uncovering the Redundancy in Transformers via a Unified Study of Layer Dropping*, TMLR.

---

> > ### Author Rebuttal · Reviewer_HAQv · 2026-04-04
> >
> > Thank you to the authors for their detailed reply. While some of my concerns are resolved. However, I remain concerned about W1 and W2.
> >
> > **W1.** I am still unclear about what the authors consider the fundamental difference between generative and non-generative tasks, and why this leads to a discrepancy in pruning performance. This point does not seem to be fully addressed in the rebuttal.
> >
> > > To directly address the decoding-length concern, we ran a controlled experiment on 48 simple arithmetic prompts (e.g., "7 + 5 ="), capping generation at 8 / 16 / 32 tokens:
> >
> > Regarding the additional experiment, I am confused about the setup. The tasks appear somewhat synthetic, and for such simple prompts (e.g., “7 + 5 =”), a small number of tokens (e.g., 8) should already be sufficient to produce the correct answer. Increasing the decoding length to 16 or 32 tokens may primarily introduce padding or irrelevant tokens rather than meaningfully testing the hypothesis. Could the authors clarify how this setup supports the intended claim?
> >
> > **Q2.**  Which pruning method do the authors use for this model?
> >
> > It is still unclear which specific pruning method is used in this work. For example, do the authors prune parameters based on magnitude values? In addition, could the authors clarify whether the pruning approach leads to improvements in inference efficiency (e.g., latency or FLOPs)?
> >
> > **W2.** scope
> >
> >  >The paper's goal is therefore primarily diagnostic: identifying where pruning remains reliable, where it breaks down, and why. This is itself a practically useful contribution: xxx
> >
> > While I understand that the paper aims to provide a diagnostic analysis of pruning behavior, the overall scope feels somewhat limited, as it mainly identifies performance differences across layers. What are the takeaways of this paper and how should practitioners adjust their pruning strategies based on these results? Should people simply avoid pruning in generative settings, or based on the inter/intra layer results, are they more efficient?

---

> > > ### Author Response · Authors · 2026-04-04
> > >
> > > Thank you for the follow-up and for clarifying the remaining concerns.
> > >
> > > ### 1. Difference between generative and non-generative tasks
> > >
> > > **This distinction is already clear at the task-definition level, and we also discuss it explicitly in the paper.** In particular, in Lines 132-189, we introduce generative and non-generative tasks separately and in detail. In Lines 196-208, we further explain that the key operational difference can be decomposed into several factors, including **iterative decoding length**, **the type of output space**, and **the dimensionality/scope of the output space**. In this sense, some non-generative tasks depend primarily on representations in the embedding spaces or on a more restricted probability subspace with a largely single-step decision, whereas generative tasks depend much more directly on the full-vocabulary probability distribution together with autoregressive decoding.
> > > **This is why the paper analyzes the discrepancy through the embedding, logit, and probability spaces, together with iterative decoding.**
> > >
> > > ### 2. Pruning effects on generative and non-generative tasks
> > >
> > > **The effects of pruning on the representation spaces associated with generative and non-generative tasks are markedly different.** Non-generative tasks remain comparatively stable in the associated representation spaces, whereas generative degradation is reflected in much larger deviation in the probability space. We present this in **Section 5**, and analyze it further in **Sections 6 and 7**, with additional results in the appendix.
> > >
> > > ### 3. Generation-Length Consideration in the Arithmetic Experiment
> > >
> > > The purpose of the arithmetic control is not to compare whether 8, 16, or 32 decoding steps make the task progressively harder, but to show that **the model can already collapse on a generative task even when the task is extremely simple and the generation length is very short.** Because 8 tokens are already sufficient for such a prompt, the severe degradation at 8 tokens is strong evidence that the failure is not explained by long generation. **Following your suggestion, we varied the decoding length** and found that severe degradation is already present at 8 tokens and remains severe at 16 / 32 as well. We also allowed some tolerance in the setup, which is why the baseline's 97.9% at @8 reflects one verbose response exceeding the token limit.
> > > This control shows that generative tasks are already severely affected by pruning even when the output sequence is very short. Thus, **the core reason for the observed performance loss is the fragility of the probability space under pruning, rather than the generation sequence being too long**; longer decoding mainly amplifies a failure that has already emerged.
> > >
> > > ### 4. Pruning methods
> > >
> > > **The pruning methods are already introduced in the main text.** In Lines 193-207, we distinguish fine-grained intra-layer pruning and coarse-grained inter-layer pruning, and adopt **Wanda** and **Layer Drop** as representative methods for these settings. The related captions and nearby paragraphs also specify the algorithm used in each experiment.
> > > **Our main focus here is the performance-side applicability of pruning, namely whether the pruned model remains reliable enough for the target task regime.** Only after this condition is satisfied does acceleration become practically meaningful. For concrete efficiency details (latency/FLOPs), we refer the reviewer to the respective pruning works [1-4].
> > >
> > > ### 5. Practical guidance
> > > **The characterization that the paper mainly identifies *performance differences across layers* does not match the scope of the paper.** More broadly, the paper systematically analyzes how pruning affects different representation spaces across task regimes, and uses this to clarify why pruning can remain reliable for many non-generative settings yet become much more fragile for generative tasks.
> > >
> > > **Our goal is therefore to characterize the applicability boundary of pruning, rather than to optimize a specific pruning recipe.** The paper is meant to explain when pruning works, when it breaks down, and why.
> > >
> > > **Accordingly, the practical implication of this applicability boundary is regime-dependent.** For non-generative tasks, pruning can still be effective and practically useful. For generative tasks, however, the paper shows that training-free pruning is much more fragile, and that adaptation after pruning is necessary to recover the degraded performance.
> > >
> > > If you have any other questions, we will be glad to address them in the coming days.
> > >
> > > Reference
> > >
> > > [1] Men et al. *ShortGPT: Layers in Large Language Models are More Redundant Than You Expect*, arXiv
> > > [2] He et al. *Uncovering the Redundancy in Transformers via a Unified Study of Layer Dropping*, TMLR
> > > [3] Sun et al. *A Simple and Effective Pruning Approach for Large Language Models*, ICLR
> > > [4] Frantar et al. *SparseGPT: Massive Language Models Can Be Accurately Pruned in One-Shot*, arXiv

---

### Decision · Program_Chairs · 2026-04-30

**Decision:**

Accept (regular)

**Comment:**

The authors study how pruning affects different parts of the LLM computation, aiming to explain why/when pruned models perform well and on what kind of tasks. The reviewers agreed that this representation-hierarchy analysis provides a novel and interesting perspective on how pruning affects. The reviewers found the experiments to be quite thorough, and noted that the authors investigate pruning affects through several different metrics, under a large variety of models and tasks.

Following the extensive rebuttal phase, the majority of technical concerns were resolved with additional experiments and clarifications provided by the authors. The key remaining concern is due to one of the reviewers being unconvinced by the explanation of the mechanisms separating generative and non-generative tasks, feeling the token-length ablation did not prove the point.

Overall, this is a technically solid, well-executed analysis paper that provides important insights into why pruning affects different tasks differently.